



# Basal melt rates and ocean circulation under the Ryder Glacier ice tongue and their response to climate warming: a high resolution modelling study

Jonathan Wiskandt[1,2], Inga Monika Koszalka[1,2,3], and Johan Nilsson[1,2]

[1]Department of Meteorology, Stockholm University, Stockholm, Sweden
[2]Bolin Centre for Climate Research, Stockholm, Sweden
[3]Stockholm University Baltic Sea Centre, Stockholm, Sweden

**Correspondence:** Jonathan Wiskandt (jonathan.wiskandt@misu.su.se)

**Abstract.** The oceanic forcing of basal melt under floating ice shelves in Greenland and Antarctica is one of the major sources of uncertainty in climate ice sheet modelling. We use a high resolution, non-hydrostatic configuration of the Massachusetts Institute of Technology general circulation model (MITgcm) to investigate basal melt rates and melt driven circulation in the Sherard Osborn Fjord under the floating tongue of Ryder Glacier, northwestern Greenland. The control model configuration, based on the first ever observational survey by *Ryder 2019 Expedition*, yielded melt rates consistent with independent satellite estimates. A protocol of model sensitivity experiments quantified the response to oceanic thermal forcing due to warming Atlantic Water, and to the buoyancy input from the subglacial discharge of surface fresh water. We found that the average basal melt rates show a nonlinear response to oceanic forcing in the lower range of ocean temperatures, while the response becomes indistinguishable from linear for higher ocean temperatures, which unifies the results from previous modelling studies of other marine terminating glaciers. The melt rate response to subglacial discharge is sublinear, consistent with other studies. The melt rates and circulation below the ice tongue exhibit a spatial pattern that is determined by the ambient density stratification.

## 1 Introduction

Increasing ice mass losses from the Greenland and Antarctic Ice Sheets result from atmosphere-cryosphere-ocean interactions, which involve a range of processes including surface ice melt, internal ice dynamics and ocean-driven basal melt, wind, tides and sea ice, often coupled in a nonlinear way (Holland et al., 2008a; Straneo et al., 2012; Smith et al., 2020; Slater and Straneo, 2022). Fresh water flux from the melting ice sheets into the ocean leads to a global sea level rise and local impacts on coastal communities worldwide, and the observed acceleration of the ice sheet melt has been attributed to anthropogenic climate change (Fox-Kemper et al., 2021). A large community effort has thus been put forward to observe, quantify and understand the underlying processes and to develop representations (parameterizations) of the ice melt processes in climate models to improve the projections of future ice sheet mass loss and its impacts (Asay-Davis et al., 2017; Edwards et al., 2014; Cowton et al., 2015; Lazeroms et al., 2018; Sheperd and Nowicki, 2017; Nowicki and Seroussi, 2018; Pelle et al., 2019). This task is





far from simple as the processes involved often feature small scales and complex geometries of both, ice and ocean, domains, and their interaction with the atmosphere.

The Greenland Ice Sheet (GrIS) holds about seven meters of sea level equivalent. It contributed 13.5 mm to the global sea level rise in the period 1992-2020, according to the most recent IPCC Report (AR6, Fox-Kemper et al., 2021). During this time there is evidence that the GrIS mass loss has accelerated in recent years (1995-2012) compared with the earlier period (Enderlin et al., 2014; Hill et al., 2018). The IPCC Report estimates a sixfold increase in mass loss rate in these last three decades from an average of 39 Gt yr$^{-1}$ in the period 1992-1999 to 243 Gt yr$^{-1}$ over the period 2010-2019 and projects the GrIS to likely contribute with 90-180 mm to sea level rise until 2100, while the Antarctic Ice Sheet contributes 30-340 mm (Fox-Kemper et al., 2021, SSP5-8.5). Ice mass loss from GrIS has a significant local fingerprint on several densely populated coastal regions worldwide (Rietbroek et al., 2016). Furthermore, freshwater input from the melting GrIS into the ocean has a potentially substantial (yet poorly quantified, and vividly debated) impacts on freshwater budget and dense water formation in the subpolar North Atlantic and hence on the strength and stability of the large scale thermohaline circulation (Rahmstorf et al., 2015; Boning et al., 2016; Luo et al., 2016; Rhein et al., 2018; Swingedouw et al., 2022). The GrIS' marine terminating glaciers drain into long and narrow fjords that connect to the open ocean. The fjords are stratified with a deeper layer of warm and saline Atlantic Water (AW), overlaid by a colder and fresher Polar Water (PW) of Arctic origin (Straneo et al., 2012). The AW enters the Nordic Seas as an upper layer of the Norwegian Atlantic Current and undergoes deepening and cooling under its poleward pathway; upon reaching the Fram Strait the AW flow bifurcates into one branch recirculating cyclonically in the Nordic Seas and the Labrador Sea, and the other one taking a detour around the Arctic Ocean (Mauritzen et al., 2011; Koszalka et al., 2013; Rudels et al., 2015). The temperature and salinity properties of AW reaching the glacial fjords around Greenland varies thus regionally. The AW that reaches the northern coast of Greenland had circulated around the Arctic Ocean and is therefore the coldest variant of AW reaching the GrIS (Straneo et al., 2012). The exposure to thermal oceanic forcing (temperature difference between the ocean water and the ice) varies therefore regionally around Greenland in addition to local differences due to wind forcing, sea ice, the mesoscale circulation on the Greenland shelf, and the fjord geometry (Seale et al., 2011; Rignot et al., 2012; Enderlin and Howat, 2013; Sciascia et al., 2013; Straneo and Cenedese, 2015; Gelderloos et al., 2017; Schaffer et al., 2017; Jakobsson et al., 2020; Wood et al., 2021).

The interactions at the glacier-ocean interface leading to a freshwater flux from the GrIS is realized through three different processes: basal melting of the submerged glacial ice, subglacial discharge (SGD) of the surface melt water (the freshwater melting at the surface ice sheet due to atmospheric forcing and percolating down through the ice and toward the ice base) during the summer, and calving of icebergs at the ice front (Straneo and Cenedese, 2015). The respective importance of the processes is dependent on the time scale and the shape of the glacier terminus. The majority of glaciers in the southern Greenland terminate as grounded, vertical ice fronts (Hill et al., 2018). These so called tidewater glaciers feature fast rising buoyant plumes, because of the steepness of the ice a the terminus (Rignot et al., 2010; Xu et al., 2012; Sciascia et al., 2013) and frequent iceberg discharge through calving. They are also subject to a relatively strong seasonal forcing due to the SGD (Sciascia et al., 2014; Straneo and Cenedese, 2015). A different type of ice-ocean interaction considers the ice shelves, i.e., the glaciers with ice tongues, found in the north of Greenland, including the Zachariae Isstrom (ZI), the Nioghalvfjerdsfjorden,





or 79°–North Glacier (79NG), the Ryder Glacier (RG) and the Petermann Glacier (PG). Floating ice tongues stabilize these glaciers by reducing the ice discharge across their grounding lines, an effect known as buttressing (Gudmundsson, 2013). On the other hand, due to the horizontal extent of the ice base, the area exposed to basal melting is much larger at ice shelves

than it is at tidewater glaciers. The observed significant inter annual variability in the grounding line position of 79NG and the observed and modelled retreat of ZI and PG have been attributed to oceanic forcing (Wilson and F. Straneo, 2015; Mayer, 2018; Choi et al., 2017; Cai et al., 2017). However, due to remoteness and logistic difficulties with the measurements, the GrIS ice shelves and their fjord outlets are still sparsely observed with regards to the ocean-driven basal melt processes.

The basal melt beneath the glacier ice tongue acts as a buoyancy source, driving a rising buoyant plume that forms an

outflow of glacially-modified water at its neutral density level. The entrainment into the plume drives an inflow of AW towards the ice base, establishing an estuarine circulation. The basal melt processes beneath ice shelves have mostly been studied in the context of Antarctic ice shelves, and have been represented in terms of a basal melt parameterization combining the basic thermodynamic considerations, conservation laws and buoyant plume dynamics, and showing a good agreement with observations (e.g. Holland et al., 2008b; Jenkins, 1991; Jenkins et al., 2010; Jenkins, 2011; Reese et al., 2018). This has guided

attempts to develop generalized versions applicable in climate models (Asay-Davis et al., 2016; Lazeroms et al., 2018; Pelle et al., 2019). However, questions remain regarding the applicability of this parameterization. One issue considers dependency of the melt on changing ambient ocean temperatures. In theory, the melt rate is linearly dependent on the temperature forcing and the boundary layer velocity, which is also linearly dependent on the temperature forcing through the buoyancy input from the melt (e.g. Jenkins, 2011; Lazeroms et al., 2018); combining to a super linear dependency of melt on temperature forcing.

Several modelling studies, however, simulate a dependency that is not significantly different from a linear one (Xu et al., 2012; Sciascia et al., 2013). Further questions consider the role of ambient ocean stratification, the ice-ocean interface geometry and the boundary layer (Holland et al., 2008b; Lazeroms et al., 2019; Bradley et al., 2021; Dansereau et al., 2013; Jordan et al., 2018). These questions are particularly relevant to the Greenland ice shelves, in addition to factors like fjord geometry, wind, sea ice, and seasonal variations of SGD. To our knowledge, there have only been few high-resolution ocean-circulation model

studies on Greenlandic ice shelves: Cai et al. (2017) investigated the sensitivity of the PG basal melt and retreat to the oceanic thermal forcing and SGD.

The third largest remaining ice tongue in North Greenland belongs to the Ryder Glacier (RG) in North Greenland (54° W, 82° N). RG terminates in the Sherard Osborn Fjord (SOF) with an ice tongue extending about 20 km from the grounding line. In contrast to the other nearby glaciers with ice tongues, RG exhibited a varied retreat and advance pattern in recent

decades (Hill et al., 2018; Wilson et al., 2017). Oceanographic surveys of SOF were completely lacking until the *Ryder 2019 Expedition* in August-September 2019 with the Swedish icebreaker Oden (Jakobsson et al., 2020). The expedition gathered a unique data set, including topographic data and hydrographic (temperature and salinity) profiles close to the ice-tongue front. The hydrographic profiles show a two-layer like stratification with a cold (about -1.5° C) and relatively fresh (salinity below 34 g kg$^{-1}$) surface layer (typical of Polar Surface Water, PSW) and a warm (0.2° C) and salty (34.7g kg$^{-1}$) layer of AW below

350 m. SOF is narrow ($\sim$ 10 km) rendering effects of the Earth's rotation negligible on the circulation, and a permanent sea-ice cover outside of SOF inhibits wind-driven water exchange between the fjord and the open ocean. The estuarine exchange



circulation in the SOF is thus driven primarily by the basal melt and the seasonal SGD flux. The weak dependence of the hydrography inside the fjord on the conditions outside distinguish RG-SOF system from the nearby glacier-fjord system at PG, and provides an interesting "laboratory" for observational and modelling studies of basal melt processes and melt-driven

buoyant flows. Furthermore, observed and modelled increases of the AW temperature in the Nordic Seas and the Arctic Ocean (Münchow et al., 2011; Straneo and Heimbach, 2013; Wang et al., 2020) rise questions of the response of the RG to increasing oceanic thermal forcing; will it respond similarly or differently to the nearby PG?

This study presents high-resolution ocean-circulation model simulations of basal melt and ocean flow in a fjord with an ice tongue. The model geometry is idealised, but its qualitative features are selected to be representative for RG and SOF. Note

that SOF has two sills, which are not represented here. This is because the present focus is on flow and melt beneath the ice tongue, which are only indirectly affected by the sills: they primarily control the features of the AW reaching the ice tongue. In control experiments, the model is initialized and, at the seaward end of the domain, restored to observations from the *Ryder 2019 Expedition* Jakobsson et al. (2020). We investigate the spatial variability of melt rates and melt driven circulation and perform sensitivity experiments to oceanic thermal forcing and SGD. In Section 2, we describe the model control configuration

and the sensitivity experiments. Section 3 presents model results from the summer and a winter control simulation and the sensitivity experiments. In Section 4, we discuss implications of the results for the future evolution of the RG and include general considerations regarding the basal melt dependence on oceanic thermal forcing and SGD.

## 2   The model

We use the MITgcm (http://mitgcm.org) that solves the Boussinesq form of the Navier–Stokes equations as a finite-difference

discretization rendered on a horizontal Arakawa C-grid, and with vertical z-levels employing partial cells (Marshall et al., 1997; Adcroft et al., 2004). The model has been used previously to study the circulation in Greenland fjords with tide water glaciers (e.g. Xu et al., 2012; Millgate et al., 2013; Sciascia et al., 2013, 2014; Carroll et al., 2015; Jordan et al., 2018) and the ice shelf-ocean interactions for Greenland and Antarctic ice shelves (e.g. Dansereau et al., 2013; Cai et al., 2017).

In our study, we consider a high-resolution, idealized, nonhydrostatic setup with a rigid lid based on the survey of Jakobsson

et al. (2020). The width of the inner fjord (ca. 9 km) is comparable to the first Rossby radius of deformation (7-10 km) which makes the across-fjord changes negligible compared to the variability along fjord (south-north) axis. Idealized three-dimensional simulations of the circulation in a SOF-like fjord with the local Coriolis parameter value confirm this notion (Yin, 2020). The rotational effects are thus neglected henceforth and the configuration is rendered two-dimensional (along fjord, vertical directions). Even at the neighbouring PG, terminating in a wider fjord of 20 km width, some previous studies

used 2D configurations, neglecting rotational effects (Cai et al., 2017). On the other hand, Millgate et al. (2013) used a 3D setup and introduced variations in the ice bathymetry (channels) in the across-fjord direction and found rotational effects on the circulation under PG. Unlike at PG, the SOF at RG is much narrower and we do not have information about the spatial variations of the ice base so we keep the 2D setup. The model parameters are listed in Table 1.



**Table 1.** Dimensional parameters used in the model simulations.

| Name | Symbol | Value | [Unit] |
|---|---|---|---|
| Specific heat capacity Ice | $c_{p,i}$ | 2000 | [J K$^{-1}$ kg$^{-1}$] |
| Specific heat capacity water | $c_{p,w}$ | 3994 | [J K$^{-1}$ kg$^{-1}$] |
| Latent heat of fusion of ice | $L_i$ | $3.34 \times 10^5$ | [J kg$^{-1}$] |
| Reference Salinity | $S_0$ | 35 | [g kg$^{-1}$] |
| Reference Temperature | $T_0$ | 0 | [$^\circ$ C] |
| thermal expansion Coefficient | $\alpha$ | $0.4 \times 10^{-4}$ | [$^\circ$ C$^{-1}$] |
| saline contraction Coefficient | $\beta$ | $8 \times 10^{-4}$ | [PSU$^{-1}$] |
| thermal/saline exchange coefficient | $\gamma_{T,S}$ | | [m s$^{-1}$] |
| thermal conductivity of ice | $\kappa_i$ | $1.54 \times 10^{-6}$ | [m$^2$ s$^{-2}$] |
| horizontal diffusivity in water (heat & salt) | $\kappa_H$ | $2.5 \times 10^{-1}$ | [m$^2$ s$^{-2}$] |
| vertical diffusivity in water (heat & salt) | $\kappa_V$ | $2 \times 10^{-5}$ | [m$^2$ s$^{-2}$] |
| Salinity coefficient of freezing temperature | $\lambda_1$ | $-5.75 \times 10^{-2}$ | [$^\circ$ C psu$^{-1}$] |
| Constant coefficient of freezing temperature | $\lambda_2$ | $9.01 \times 10^{-2}$ | [$^\circ$ C] |
| Pressure coefficient of freezing temperature | $\lambda_3$ | $-7.61 \times 10^{-8}$ | [$^\circ$ C Pa$^-$1] |
| reference Density | $\rho_0$ | 999.8 | [kg m$^{-3}$] |
| horizontal viscosity | $\nu_h$ | $2.5 \times 10^{-1}$ | [m$^2$ s$^{-2}$] |
| vertical viscosity | $\nu_v$ | $1 \times 10^{-3}$ | [m$^2$ s$^{-2}$] |

The domain's dimensions and geometry are shown in figure 1a and b. We focus on the circulation in the ice shelf cavity,
i.e., the first 30 km of the SOF with a horizontal grid spacing of $dx = 10$ m along the fjord axis. The model width in the
across-fjord direction is one grid cell of size $dy = 10$ m. The domain is 1,000 m deep divided in 300 equally-spaced vertical
levels ($dz = 3,33$ m). The first 20 km of the domain are covered by a floating ice shelf representing the RGs ice tongue. The
ice tongue terminates in a 50 m deep front at $x = 20$ km. To represent the observations, the ice base is set to be a constant
linear slope of s = 0.045, which is equivalent to an angle of $\phi = 0.045^\circ$ , connecting the grounding line and the lowest point of
the calving front (Fig. 1a). The grounding line is set to 50 m above the ocean floor to avoid instability issues at the corner and
leave a space for the plume to develop (Burchard et al., 2022). The bottom of the domain is flat. A quadratic drag is applied at
the bottom of the domain and the ice.

All experiments are started from rest, initialized with horizontally uniform salinity (S) and temperature (T) profiles. In the
control simulations these approximate the hydrographic profiles taken glacier ward of the inner sill just in front of the ice front
(Station 16, 17 from figure 1 in Jakobsson et al. (2020)). We set up a winter control simulation (*control_win*) without any
subglacial discharge and a summer control simulation with subglacial discharge (*control_sum*). For simplicity and because the

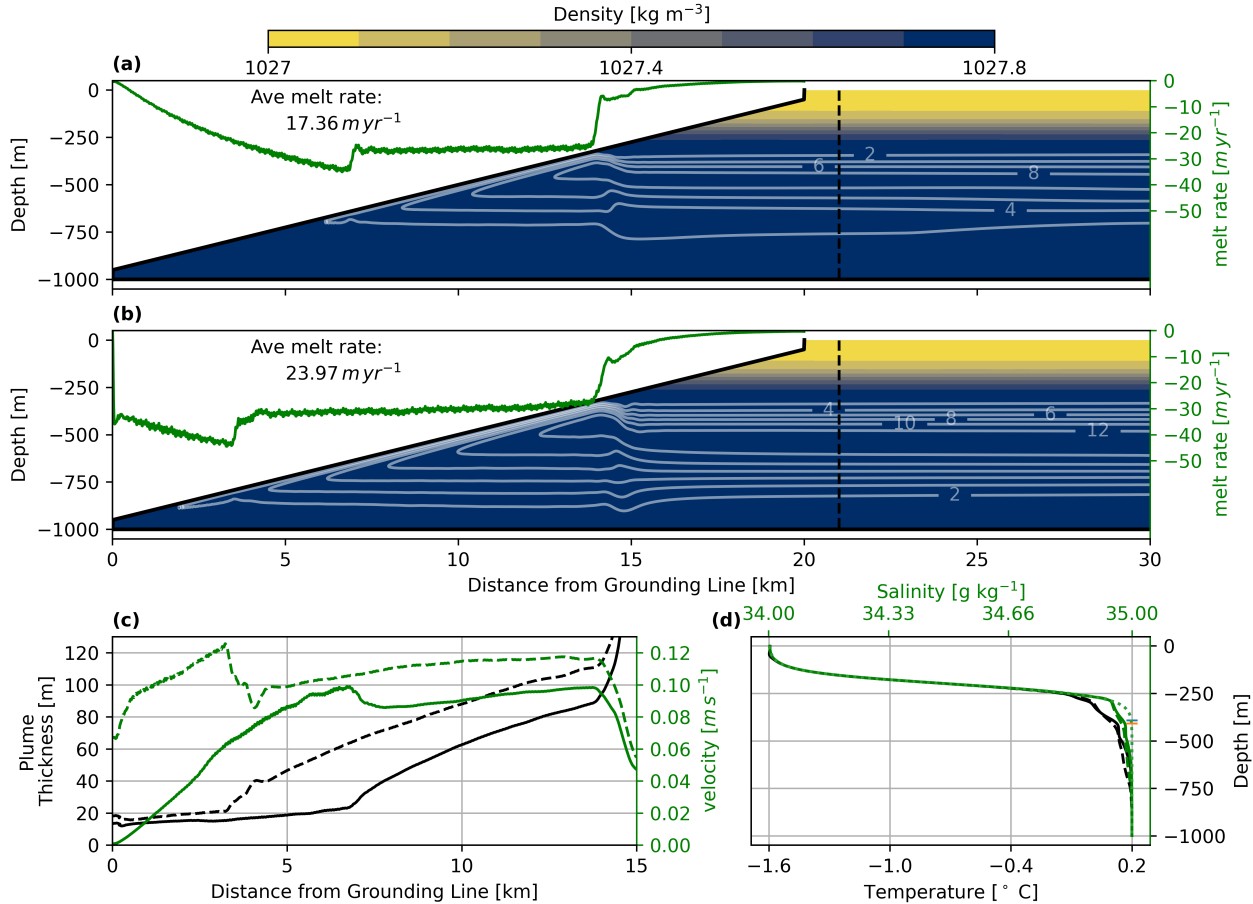

**Figure 1.** a) The stream function (white contours in m$^2$ s$^{-1}$) of the steady circulation superimposed on the density ($\sigma$, colors) and the melt rate (green line, right axis) along the ice ocean interface (black line) for *control_win*. The black dashed line indicates the location of profiles shown in figure 2 and 6; b) same as in a) but for *control_sum*; c) The plume thickness (black) calculated for summer (dashed) and winter (solid) control simulation; and the vertically averaged plume velocity (green) for summer (dashed) and winter (solid) control simulations. d) Initial and open ocean boundary condition profiles of salinity and temperature (showing as one blue dotted line for the chosen axes limits) and the steady state temperature (black) and salinity (green) profiles of the summer (dashed) and winter (solid) control simulations at $x = 21$ km.

nonlinear effects are small in the range of S-T values we are considering a linear equation of state for the density $\rho$:

$$\rho = \rho_0 \big[ 1 - \alpha(T - T_0) + \beta(S - S_0) \big], \tag{1}$$

with parameters listed in Table 1. Sub grid scale processes are parameterized using a Laplacian eddy diffusion of temperature, 
salinity, and momentum with constant coefficients as in the MITgcm fjord simulation of comparable resolution by Sciascia





et al. (2013). At the model resolution, the mixing processes are dominated by turbulence, so we apply equal values of diffusion coefficients for all variables (Table 1).

The northern boarder of the fjord (at $x = 32$ km) is the only open boundary. The outflow is balanced at the boundary yielding a net zero cross boundary flow. Temperature and salinity are restored to the initial conditions in a 2 km wide restoring zone with a restoring timescale of one day at the innermost grid point ($x = 30$ km) and one hour at the outermost point ($x = 32$ km). An experiment conducted in a horizontally extended domain (not shown here) shows, that the boundary is sufficiently far away from the ice to have negligible effects on the evolution of the circulation underneath the ice tongue.

## 2.1 Basal melt parameterization

To parameterize the basal melt processes at the RG's ice shelf, we use the SHELFICE package[1] (Losch, 2008) applying ice ocean interactions in an interface mixed layer, defined as the uppermost grid cell adjacent to the ice ocean interface (Dansereau et al., 2013; Jordan et al., 2018). Freezing and melting processes occur at the infinitesimal boundary layer at the interface and are paramaterized employing the three-equation formulation (Hellmer and Olbers, 1989; Holland and Jenkins, 1999):

$$T_b = \lambda_1 S_b + \lambda_2 + \lambda_3 P_b \tag{2}$$

$$c_{p,w} \rho_i \gamma_T (T_w - T_b) = -L_i q - \rho_i c_{p,i} \kappa_i \frac{(T_s - T_b)}{H_i} \tag{3}$$

$$\rho_i \gamma_S (S_w - S_b) = -S_b q \tag{4}$$

The interface boundary layer temperature ($T_b$) is the in-situ freezing point temperature obtained from the boundary layer pressure and salinity ($P_b$ and $S_b$ respectively) using the linear equation of state (Eq. 1) where $\lambda_j$ are constants. Equations 3 and 4, that describe heat and salt balances at the interface, respectively, are used to calculate $S_b$. We assume a linear temperature profile in the ice and approximating the vertical temperature gradient in the ice as the difference between the ice surface ($T_S = -20°$ C) and interface (ice bottom) temperatures ($T_i$) divided by the local ice thickness. Subscript $w$ refer to the properties in the interface mixed layer. The values of parameters are listed in Table 1.

Exchange coefficients for salt and heat are calculated online (Holland and Jenkins, 1999) based on the along ice boundary layer velocity $u^* = c_D \sqrt{u_{BL}^2 * w_{BL}^2}$, where $c_D$ is the models drag coefficient and $u_{BL}$ and $w_{BL}$ are the local horizontal and vertical boundary layer averaged velocities. This yields:

$$\gamma_{T,S} = \frac{u^*}{\Gamma_{Turb} + \Gamma_{Mole}^{T,S}} \tag{5}$$

where $\Gamma_{Turb}$ and $\Gamma_{Mole}^{T,S}$ are the turbulent and molecular exchange parameters defined as in Holland and Jenkins (1999) equations (15) and (16). The linear dependency of the exchange coefficient on the along-ice velocity $u^*$ is expected to lead to a super-linear dependency of melt on the temperature forcing, because $u^*$ is approximated to be increase with increasing temperature forcing through the change in buoyancy from enhanced melting (e.g. Jenkins, 1991; Holland et al., 2008a; Jenkins, 2011; Lazeroms et al., 2018).

---

[1]https://mitgcm.readthedocs.io/en/latest/phys_pkgs/shelfice.html





Equations 2-4 are solved for boundary temperature and salinity and the melt rate $q$ at every time step. The fresh water mass flux output (in kilograms per square meter and second [kg m$^{-2}$ s$^{-1)}$]) is negative for melting, i.e., a downward mass input into the ocean. The temperature and salinity changes due to fresh water flux are implemented using virtual fluxes in the respective tendency equations. As the model employs partially filled cells, the parametrization uses a simple boundary layer averaging over vertical grid size $dz$. Velocities are averaged onto the tracer grid points. For further details about the ice shelf parametrization the interested reader is referred to Losch (2008).

## 2.2  Sensitivity experiments

*Oceanic thermal forcing*

First, we investigate a scenario of warming AW temperatures. To this end, we conduct a set of experiments with varying AW temperature ($T_{AW}$) applied as initial condition and boundary condition at the open ocean boundary. We define a temperature forcing (TF= $T_{GL} - T_b$) where $T_{GL}$ is the water temperature at the grounding line and $T_b$ is the freezing point temperature evaluated at the grounding line depth using the local salinity ($S_b$) and quantify the response of the system in terms of the melt rate and circulation changes to changing TF. We apply a wide range of AW temperatures to quantify the response of the melt rate and the resulting circulation to varying TF with more confidence.

*Subglacial discharge*

A second set of sensitivity experiments is conducted to investigate the influence of subglacial discharge (SGD). Due to a lack of accurate estimates, the SGD volume fluxes are set to fractions of the integrated melt flux of the winter control simulation. SGD is implemented by relaxing the values of temperature, salinity and horizontal velocity at the grounding line towards the local freezing point temperature, zero salinity and a discharge velocity calculated based on the discharge volume in the temperature and salinity tendency equations.

*Steady state*

All simulations were run for 100 days with a time step of $10s$ for the control runs and varying time steps for sensitivity experiments (Table 2). The statistically stationary equilibrium is reached after ca. 40 days for volume-averaged kinetic energy, circulation time scales and melt rates for all the runs (Figure B1 and B2), which is in line with an overturning time scale of 20-30 days. The integrated temperature change does not stabilize completely (Figure B1) for the two warmest runs but the deviations do not have significant effect on the other properties. For further analysis we use the last 40 days of simulation (model days 60-100). The experiment setup details and key diagnostic values for a selected subset of experiments is given in Table 2. For the complete list of experiments we refer the reader to section A.





**Table 2.** Setup parameters and diagnostics for selected experiments. From left to right: AW temperature, subglacial discharge volume in percent of *control_win* integrated melt volume, model time step, temperature forcing, overturning time scale, averaged melt rate/ ice retreat, integrated melt flux per unit width in transverse direction. For a complete account of all experiments see section A.

| ExpName | $T_{AW}$ | SGD | dt | TF | $\tau_o$ | Ave. Melt | Melt Flux |
|---|---|---|---|---|---|---|---|
| | [° C] | [%] | [s] | [° C] | [days] | [m yr$^{-1}$] | [km$^3$ yr$^{-1}$] |
| nAW20 | -2.0 | 0 | 10 | 0.68 | 78 | 0.92 | 0.16 |
| AW00 | -0.0 | 0 | 10 | 2.68 | 27 | 15.28 | 2.60 |
| **control_win** | 0.2 | 0 | 10 | 2.88 | 27 | 17.36 | 2.95 |
| AW20 | 2.0 | 0 | 10 | 4.68 | 23 | 37.43 | 6.36 |
| AW40 | 4.0 | 0 | 5 | 6.67 | 22 | 61.34 | 10.43 |
| AW60 | 6.0 | 0 | 5 | 8.67 | 22 | 83.91 | 14.27 |
| **control_sum** | 0.2 | 10 | 5 | 2.88 | 18 | 23.96 | 4.07 |
| sgd020_AW02 | 0.2 | 20 | 5 | 2.88 | 15 | 26.67 | 4.54 |
| sgd050_AW02 | 0.2 | 50 | 5 | 2.87 | 12 | 31.60 | 5.38 |
| sgd100_AW02 | 0.2 | 100 | 3 | 2.86 | 10 | 36.67 | 6.24 |
| sgd010_AW20 | 2.0 | 10 | 5 | 4.67 | 17 | 47.96 | 8.16 |
| sgd010_AW40 | 4.0 | 10 | 5 | 6.67 | 16 | 76.59 | 13.03 |

# 3 Results

## 3.1 Winter and summer control simulations

The steady state (model days 61-100) melt rates and circulation under the RG ice tongue for *control_win* and *control_sum* simulations are shown in figure 1a and b, respectively. Both cases exhibit an estuarine circulation typical of glacial fjords (Straneo and Cenedese, 2015): the warm AW inflow in the lower layer supplies heat to the ice base forcing basal melting. The melt water input drives a buoyant plume, which rises into the base of the pycnocline (located at about 400 m depth) where it reaches its level of neutral buoyancy and forms a horizontal outflow jet towards the open boundary. The overturning time is estimated from the model domain volume ($V_d$) divided by the integrated AW volume transport at $x = 21$ km ($\tau_O = \frac{V_d}{\int \int u_{AW}(z)\,dzdy}$) and yields 27 days (winter) and 18 days (summer, Table 2).

Restoring to the initial stratification at the open boundary results in a continuous oceanic heat transport toward the ice base sustaining the basal melt (Eqs. (2) - (4)). The steady state melt rates along the ice base are shown in figure 1a and b, and the average values are shown in table 2. Both, winter and summer control simulations, exhibit negative average melt rates, corresponding to equivalent ice thickness loss and glacier retreat. In *control_win*, the average melt rate is 17.36 m yr$^{-1}$ but the melt rates are variable along the ice base (Figure 1a and b): rising from zero at the GL to a maximum of 35.08 m yr$^{-1}$ at about 7 km where they drop slightly to a value around 27 m yr$^{-1}$ persisting until 14 km, and then dropping to zero. This spatial melt



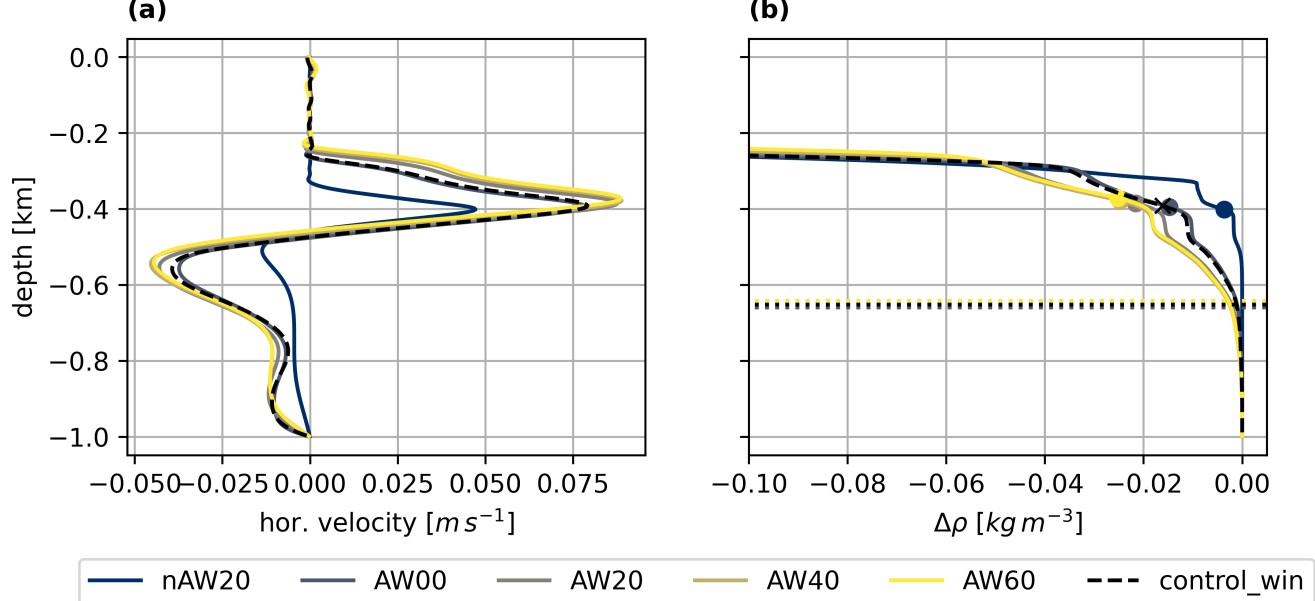

**Figure 2.** Profiles (solid lines) at 21 km from the *control_win* and selected temperature forcing experiments of (a) horizontal velocity and (b) density change with respect to bottom density, $\Delta\rho = \rho(z) - \rho(z = 1km)$. Dots in (b) indicate the depth of maximum horizontal velocity. The horizontal lines in (b) indicate the depth of maximum melt (corresponding to the plume's regime transition point depth).

rate distribution is related to the buoyant plume properties (see below). The melt water flux integrated along the 20 km long ice
shelf amounts to $3.47 \times 10^5$ m$^2$ yr$^{-1}$ per unit width, or 2.95–3.47 km$^3$ yr$^{-1}$ for the estimated glacier tongue width of 8.5–10 km. For the control summer simulation, the average basal melt increases to 23.96 m yr$^{-1}$ (or 4.07–4.79 km$^3$ yr$^{-1}$), which is an increase of 38% compared to the winter control. The summer control shows a similar variability of melt rates along the ice base to the winter control but for the immediate buoyancy input at the GL, which leads to the melt rate maximum shifting the transition zone from 7 km to closer to the GL at 4 km where the maximum melt rate is 44.50 m yr$^{-1}$, and a subsequent drop to
an approximately constant 30 m yr$^{-1}$ persisting until 14 km, and then dropping to zero. This shift of transition zone (7 km in winter vs. 4 km in summer) collocates with a downward thickening of the ambient pycnocline (Figure 1d).

We will here describe the melt driven circulation for the winter simulation, and examine effects of changes of thermal forcing and SGD in the following sections. To characterize the buoyant plume, we define the plume as the region beneath the ice base where $u > 0$ (the flow is towards the open ocean). We tried alternative definitions of the plume based on the temperature and
225 salinity difference compared to the ambient and prescribed stratification. These resulted in a wider plume over the distance between 7 and 14 km. As the difference encompasses the region of no horizontal flow outside the plume (by definition $u \leq 0$ here), this has no impact on the further calculations of e.g., plume transport and we decide to stick with the definition based on the horizontal velocity.





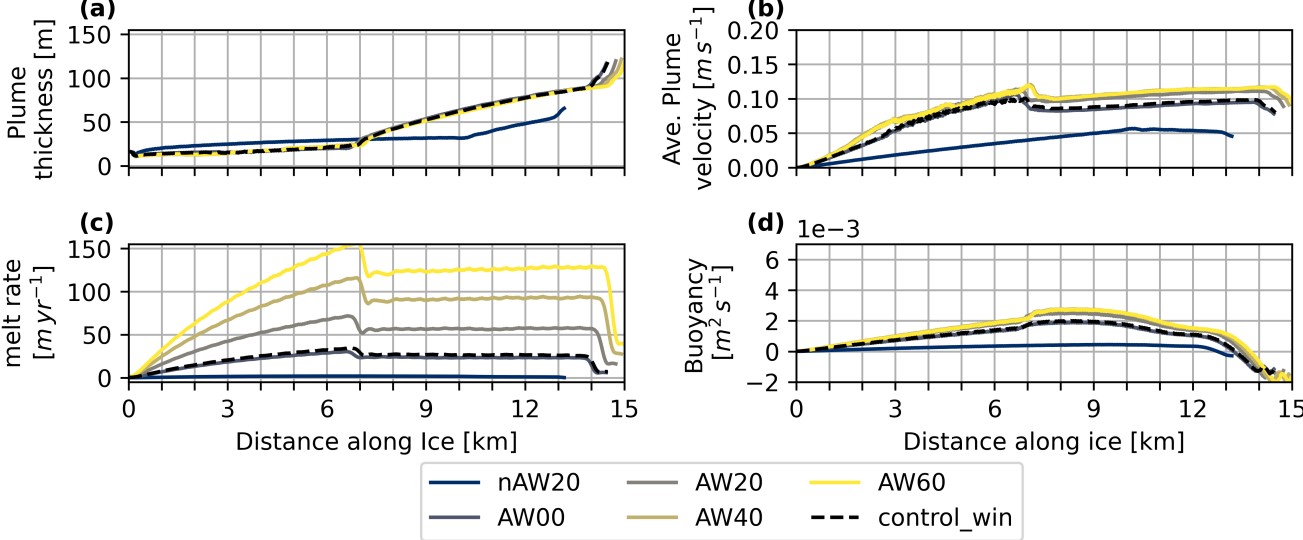

**Figure 3.** Plume properties for simulations with varying oceanic thermal forcing (AW temperatures) as a function of distance from the grounding line along the ice: (a) plume thickness, (b) averaged plume velocity, (c) melt rate and (d) Buoyancy (see text).

The plume thickness and averaged plume velocity ($u_p = \sqrt{u^2 + w^2}$) are shown in figure 1c. Clearly distinguishable are
230 two different plume regimes during its ascent along the ice base: the accelerating plume and the thickening plume. In the accelerating plume regime close to the GL, the plume has a thickness of around 20 m, while the average plume velocity increases steadily to a maximum of 0.1 m s$^{-1}$ at 7 km. In the thickening regime the velocity is around 0.095 m s$^{-1}$ and the plume thickness increases from 20 m to 90 m between 7 km and 14 km. This two-regime structure is evident in other plume properties (e.g., temperature, salinity and density; not shown) and is corresponding to the spatial variability in the melt rates
described above. The depth of the transition from accelerating to thickening plume is linked to the ambient stratification in the fjord (Figure 1d, see Section 3.2 and 3.3).

At 14 km, the plume velocity drops to zero (Figure 1c) which marks the location where the plume separates from the ice (Figure 1a,b) and forms a horizontal outflow jet towards the open boundary. The outflow layer is about 250 m thick (spanning 250–500 m depth) with a maximum velocity at 400 m (Figure 2a). The outflow forms a T-S transition layer between the AW
and the PW, that was smoothed out in the idealized initial profiles (Figure 1d and 2b). This transition is recognizable in the observations of (Jakobsson et al., 2020), lending confidence to the model results. The outflow at intermediate depth is balanced by an AW inflow in the bottom layer with a maximum velocity of -0.04 m s$^{-1}$ just below 500 m and a secondary maximum close to the bottom (Figure 2a). The plume is not sufficiently buoyant to penetrate into the upper layer of PW which remains undisturbed.





## 3.2 Sensitivity to oceanic thermal forcing

We will first describe the results of the winter simulation without SGD for different temperature scenarios, before looking into the effect of the varying SGD (Sect. 3.3). We applied a wide range of AW temperatures to quantify the response of the melt rate and the resulting circulation to varying TF with more confidence. The response of the melt driven circulation to changing temperature forcing (TF) is shown in figures 2 and 3. The structure of the circulation and the distribution of the plume properties is the same for all experiments, except of those with very low AW temperatures (TF$< 2°$ C, $T_{AW} < -1.0°$ C). The plume thickness and its velocity (Figure 3a and b), thus the volume transport, change only slightly in response to the increased melt for warmer experiments (Figure 3c). The increased melt water input freshens and cools the plume and the outflow, sharpening the pycnocline in the outflow without changing its thickness (Figure 2b). Figure 3d shows the buoyancy in the plume, estimated as the density difference between the local plume density and the ambient ocean at 21 km: $b = (\rho_a(x = 21km, z) - \rho_p(x, z)) g$. Because of the competing effect of freshening and cooling on the density, there is no effective change of buoyancy forcing with increasing TF. For the coldest experiments, i.e., weak oceanic thermal forcing, the melt rate is lower and the plume does not develop the two-regime structure we see in warmer experiments.

The horizontal dashed lines in figure 2b show the depth of maximum melt rates corresponding to the plume transition between the accelerating and thickening (Section 3.1) with respect to the ambient stratification. For all experiments the depth of the transition coincides with the base of the pycnocline marked by $\Delta\rho < 0$ (at about 620 m depth). This suggests that the spatial structure in the melt rates and the transition between the accelerating and thickening plume at 7 km is determined by the ambient stratification. The evolution of the vertically averaged plume buoyancy along the ice underpins this conclusion further, as the maximum buoyancy coincides with the point of regime transition for various TF experiments (Figure 3d).

Figure 4a shows the average melt rate for a wide range of oceanic thermal forcing (TF). We quantify the response to oceanic thermal forcing using regression analysis (e.g., Storch and Zwiers, 1984) and a resampling technique. A linear regression fit has high residuals for low TF values. We then construct sample subsets by successively excluding data points from cold experiments, starting with the coldest, and re-evaluate the linear fit. In doing so, we find the highest coefficient of determination ($R^2$) and the lowest root mean squared error of a linear fit for experiments with a temperature forcing larger then the cut-off value $2.88 <$TF$_c < 3.18°$ C (Figure 4). The adjusted linear fit has smaller residuals across the whole TF range (Figure 4a) implying a non-linear dependency of melt flux on TF for TF$\leq 2.88°$ C and a linear dependency for TF$\geq 3.18°$ C. The fitted linear increase of melt per degree warming of AW is 11.69 m yr$^{-1}$ K$^{-1}$ or roughly two thirds of the modelled melt under winter conditions (17.36 m yr$^{-1}$) per degree warming.

The integrated cooling and freshening effect on the plume's buoyancy is summarized for all temperature sensitivity experiments in figure 4b. The buoyancy due to the plume temperature (Buo-T) and salinity (Buo-S) is calculated as the buoyancy in figure 3 but from the respective difference between temperature and salinity using the linear equation of state (Equation 1) and integrated vertically and horizontally over the plume. For higher temperature forcing (TF$\geq$TF$_c$, Table A1) the buoyancy is not longer increasing linearly with TF. The effect of temperature and salinity start to balance one another and the total buoyancy becomes independent of temperature for experiments with thermal forcing of TF$> 6.18°$ C. This explains the very weak re-





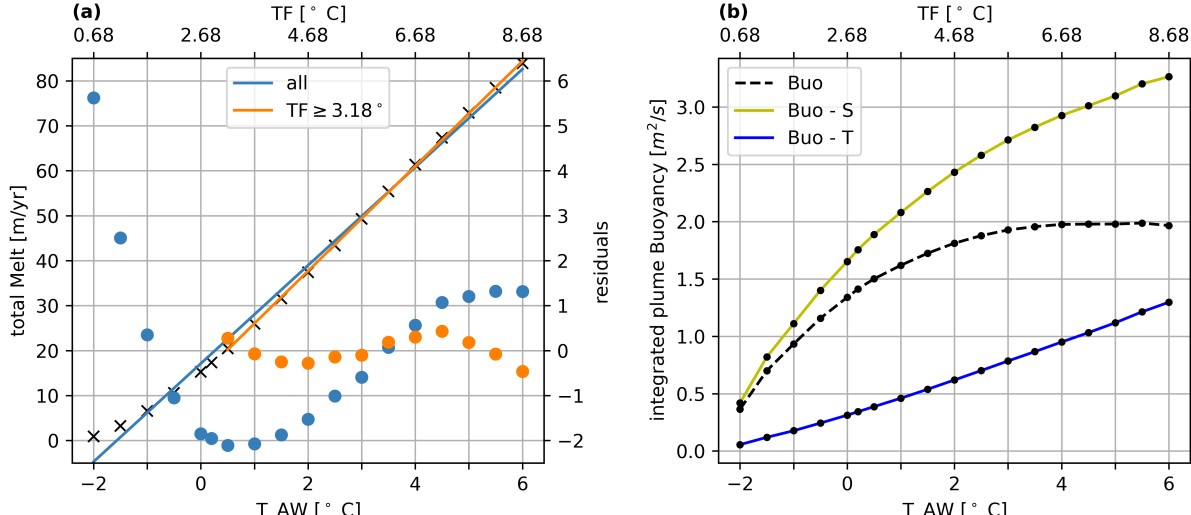

**Figure 4.** (a) The average melt (left ordinate axis) as a function of AW temperatures ($T_{AW}$; bottom abscissa) corresponding to thermal forcing (TF; top abscissa) for winter experiments (without subglacial discharge). Superimposed are the linear fit for all experiments (blue line) and for intermediate to warm experiments only (orange line; see text). The corresponding residuals (right ordinate axis) are plotted with dots. (b) The plume averaged buoyancy due to temperature (Buo-T; blue; absolute values of the negative function are shown), salinity (Buo-S; yellow) and the combined influence on density (Buo, black dashed).

sponse of plume velocity to the oceanic thermal forcing at higher TF (Sect. 3.1). Consistently, the fjord overturning time scale
decreases with TF for colder experiments (implying a faster overturning) but saturates around $22 - 23$ days for the warmer
simulations (Table 2 and A1).

## 3.3 Sensitivity to subglacial discharge

Subglacial discharge (SGD) has a pronounced effect on the basal melt rates. The average melt rate for the *control_sum* simulations (where SGD is set to 10% of the average basal melt flux for the control winter; Table 2), is increased by 38% (from 17.36
m yr$^{-1}$ to 23.96 m yr$^{-1}$, Table 2). For the experiment with the highest SGD (*sgd100_AW02* in table 2) the increase in melt is
111% (36.67 m yr$^{-1}$).

Not only does the total melt change, but so does the melt rate distribution along the ice base and the plume properties (Figure 5). The buoyancy input from SGD lead to high plume velocities at the GL resulting in higher melt rates there (Figure 5a-b). While for all experiments the accelerating and thickening plume regime identified in *control_win* are distinguishable
by thickness, velocity and melt (Figure 5a-c), the point of transition moves towards the GL. For *control_sum*, *sgd010_AW20* and *sgd010_AW_40* the transition point jumps more than 3 km closer to the GL (from 7 km in *control_win* to 3.5–4 km in *control_sum*). When increasing the discharge further, the migration of the point of transition towards the GL becomes less rapid (to 3–3.5 km for 20% discharge <3 km for 50% discharge). This does not immediately reflect in a thickening of the



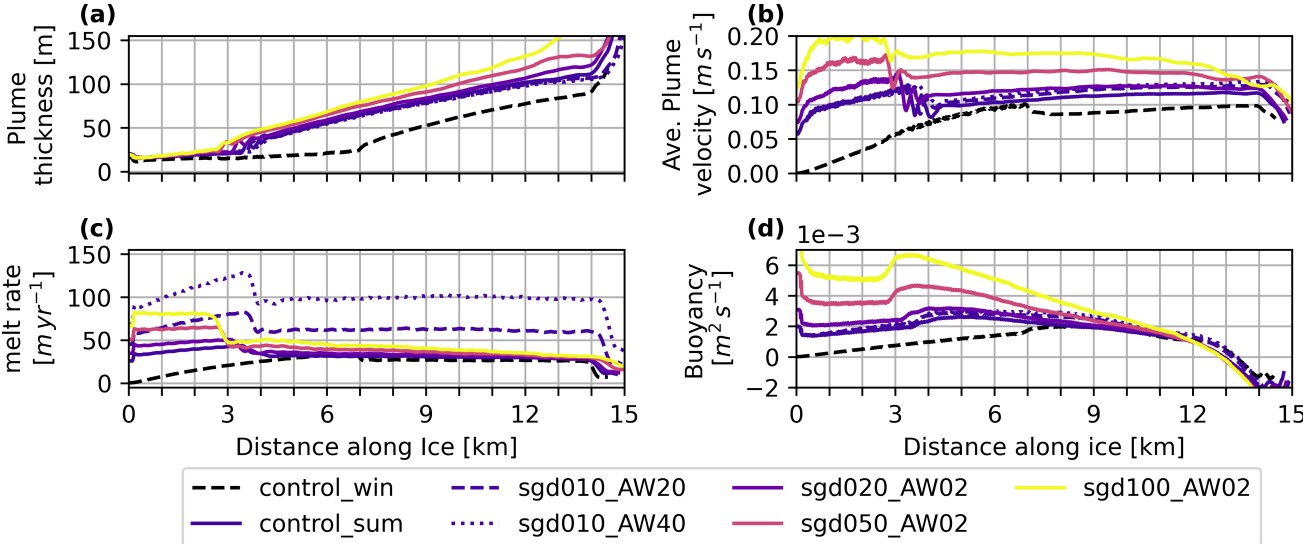

**Figure 5.** Same as Figure 3 but for different SGD volume fluxes and AW temperatures.

plume (Figure 5b), which is only slightly increased compared to the *control_win*. Despite starting with already high velocities,

the plume does accelerate further in the first regime, while the melt rate increases and the thickness stays constant, similar to the winter simulations. In the thickening regime, after a slow down of the plume, the velocities and melt rates become virtually constant while the plume continues to thicken (Figure 5.

The increased melt water input in simulations with SGD leads to a fresher, colder and faster outflow and a downward shift of the base of the pycnocline (Figure 6a-b), more pronounced for experiments with higher SGD. This downward shift of the base

of the pycnocline to a depth of about 800 m is related to the spatial structure of the melt rates and the shift of transition zone between the accelerating and thickening plume regimes (Figure 5a-c and Figure 6b; horizontal dotted lines), consistent with findings in Sect. 3.2 (Figure 2). The distribution of the plume buoyancy along the ice base underpins this conclusion further, as the maximum buoyancy coincides with the point of regime transition for all SGD experiments (Figure 5d).

The effect of oceanic thermal forcing (increasing TF) on simulations with subglacial discharge is shown in figure 7. It leads

to the following observations: i) the functional response of the melt rate to TF found in the winter simulations (without SGD; Figure 4a) holds for the simulations with SGD (Figure 7a), ii) there is stronger linear increase in the melt rate with TF for experiments with SGD as compared to the experiments without SGD (Figure 7a), iii) for experiments with constant TF, the melt rates increases less than linear (in a fractional manner) with the SGD (Figure 7b).




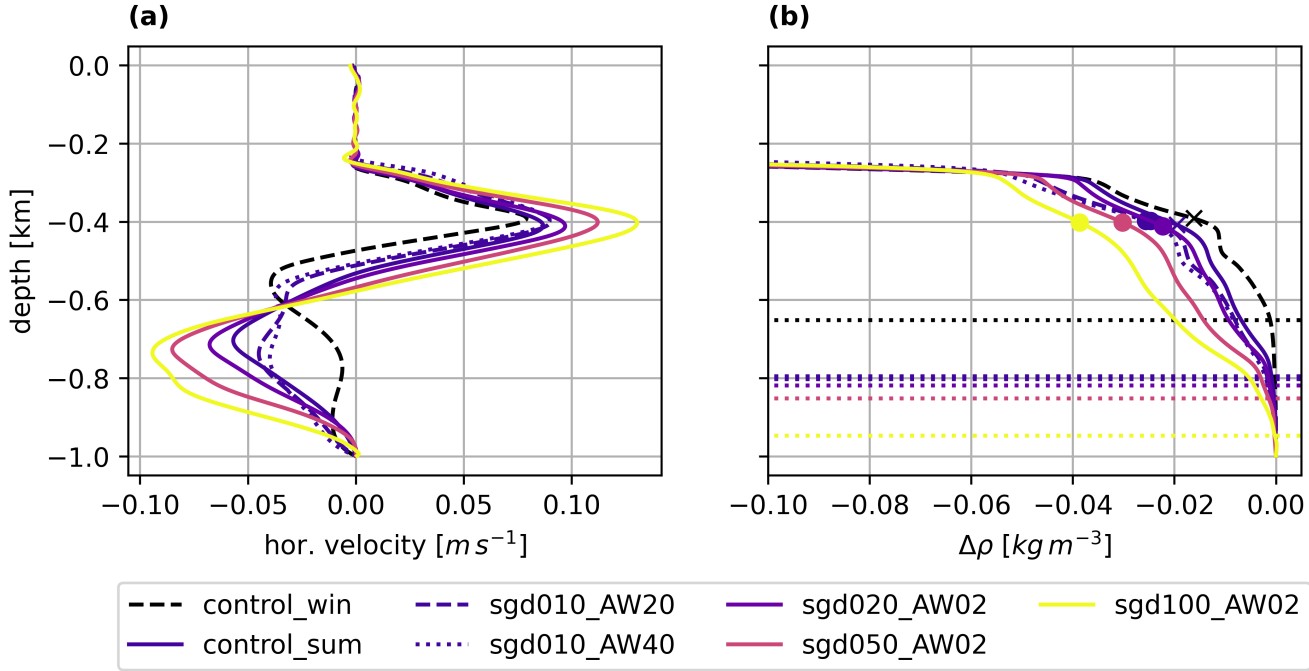

**Figure 6.** As Fig. 2 but for different SGD volume fluxes.

## 4 Discussion and conclusions

We used a high resolution, non-hydrostatic configuration of the MITgcm to investigate basal melt rates and melt driven circulation in a fjord with an ice tongue. The fjord–ice-tongue geometry is highly idealized, but the grounding-line depth and ice-tongue length are selected to represent Ryder Glacier in Sherard Osborn Fjord, northwestern Greenland. The basal geometry of Ryder's ice tongue varies across the fjord, a feature that cannot be represented in the present two-dimensional model. For simplicity, we have chosen an ice-tongue with a linear basal slope, which roughly corresponds to the area-averaged basal

slope of Ryder. The control model configuration is based on the observational survey of the *Ryder 2019 Expedition* and, to our knowledge, our study is the first to investigate aspects of this glacier–fjord system using high-resolution ocean modelling. A protocol of model sensitivity experiments quantified the response to oceanic thermal forcing due to warming Atlantic Water (AW), and to the buoyancy input from the subglacial discharge (SGD) of surface fresh water. We applied broad ranges of varying AW temperatures and SDG fluxes to better resolve the basal melt response to forcing and to make our model experiment

more universal and relevant to future development of basal melt parameterizations in climate ice sheet models.



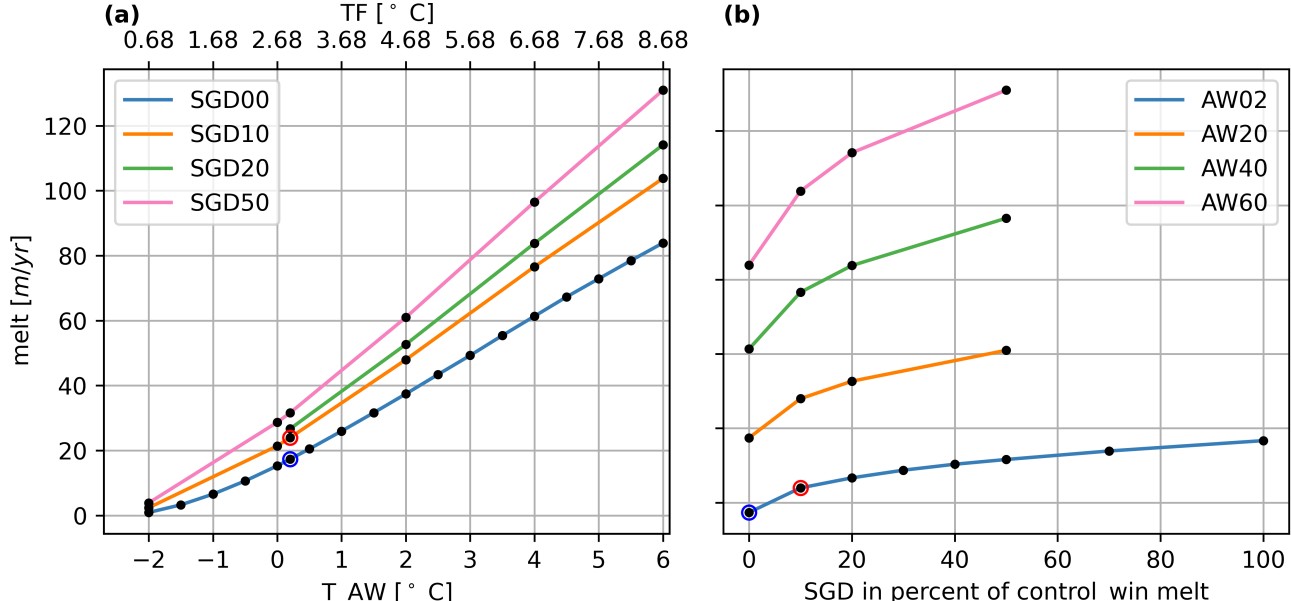

**Figure 7.** (a) The average melt (left ordinate axis) as a function of AW temperatures ($T_{AW}$; bottom abscissa) corresponding to thermal forcing (TF; top abscissa) for summer experiments for summer model experiments with added SDG (dots). The colored lines link model simulations with equal SGD. (b) The average melt as a function of SGD (dots). The colored lines indicate sets of experiments with equal thermal forcing. The blue and red circles indicate winter and summer control simulations, respectively.

*Model representation of the glacier-fjord system*

Our control simulations represent salient features of estuarine circulation typical of Greenlandic glacier–fjord systems subject to oceanic thermal forcing due to the AW inflow (Straneo and Cenedese, 2015): the warm AW inflow in the deeper layer supplies heat to the ice base forcing basal melting. The melt water is fresher than ambient and drives a buoyant plume underneath

the ice tongue. The plume rises into the base of the pycnocline where it reaches its level of neutral buoyancy, detaches from the glacier front, and intrudes horizontally into the ambient water forming an outflow jet back towards the open boundary. The entrainment of ambient water in the rising buoyant plume drives a slow flow of ambient waters toward the glacier.

The simulated melt rates for our idealized Ryder Ice Tongue, which has a linear basal slope, are broadly comparable to the satellite-derived estimate from the real Ryder for 2011–2015 by Wilson et al. (2017): our maximum melt rates near the

grounding line are 40–50 m yr$^{-1}$ while they are slightly higher (around 20 m yr$^{-1}$) away from the grounding line compared to the observed (10–20 m yr$^{-1}$). The area-integrated basal melt for the control winter experiment (taking the ice tongue width of 8.5 km) is about 3 km$^3$ yr$^{-1}$ as compared to the observed 1.8±0.21 km$^3$ yr$^{-1}$ (Wilson et al., 2017), while it is higher, about 4 km$^3$ yr$^{-1}$, for the summer control experiment. The simulated steady state fjord stratification recovers the observed signature



of an outflow of glacially-modified water, which was smoothed out in the profiles used for initialization, providing additional
qualitative support for the feasability of our model approach.

*Spatial structure of basal melt rates and melt driven circulation*

Our high resolution model simulation allowed to resolve a spatial pattern of the basal melt and the melt driven circulation under
the ice tongue. In the winter control simulation, the basal melt rates and the plume exhibit a two-regime structure along the ice
base (high melt rates in the accelerating plume regime up to 7 km and the lower melt rates in thickening plume regime thereafter
up to 14 km). To our knowledge, small-scale variations in the melt rate have been barely captured by observations (Wilson
et al., 2017). The two-regime structure persists in the sensitivity model runs with varying ocean thermal forcing. Applying
an additional buoyancy source in simulations with SGD shifts the transition between the two regimes closer to the grounding
line. Our results suggest that this spatial structure of the basal melt rates and the melt driven circulation is determined by the
ambient density stratification as shifts of the transition zone in various sensitivity experiments relates to the downward shift
of the pycnocline and shift in buoyancy forcing. Notably, in the first regime close to the grounding line, our simulated melt
rates in the winter runs (without SGD) show a monotonic increase rather than a broader maximum found in Petermann Glacier
simulations of Cai et al. (2017). This monotonic increase is less pronounced in our simulations with SGD (SGD was applied
in their study) but it could also be attributed to different ice geometry (a steep ice base close to the grounding line in their
study, which would lead to increased melt rates there). Other factors that could affect the structure of the melt rates, but are
unresolved by either modelling study, are the variability of the SGD in the transverse direction as it enters the fjord waters
through channels discharging at the base of the glacier's front whose number, sizes, and geometries and time variability are
mostly unknown and possibly influenced by the complex networks of drainage channels and crevasses in the glaciers (Chen,
2014) and the presence of basal channels and terraces (Millgate et al., 2013; Dutrieux et al., 2014).

*Response to oceanic thermal forcing*

In this study, we investigated the response of the melt rates and melt driven circulation to the oceanic thermal forcing, TF
(varying AW temperatures). The form of the applied basal melt parametrization (Eqs. 2-3) suggests a non-linear dependence
of the basal melt on TF, since the melt rate depends on both, the ocean temperature and the plume velocity through the transfer
coefficient (Eq. 5). The plume velocity is in turn dependent on TF through the buoyancy input from the melt (Holland et al.,
2008a; Jenkins, 2011; Lazeroms et al., 2018). A nonlinear relation was found in former studies of Antarctic ice shelves subject
to ocean water temperatures around zero degrees (Holland et al., 2008b). On the other hand, several modelling studies of
tidewater glaciers around Greenland, where ocean temperatures are higher due to the AW inflow, have reported on a linear
dependency of melt rates on TF (Xu et al., 2012; Sciascia et al., 2013, 2014). A modelling study of Petermann Glacier, a
neighbour of Ryder Glacier, by Cai et al. (2017) found a slightly non-linear dependency of melt on TF using a similar set of
sensitivity experiments as presented here and assuming same relationship for the whole TF range. Here, we applied a wide range
of oceanic thermal forcing and a resampling technique to quantify the response of the melt rate and the resulting circulation to




varying TF in more detail. We found that a non-linear relationship holds for the simulations with low TF (TF≤ 2.88° C, Figure 4a), while it becomes linear for higher TF, thus unifying results from the previous studies.

We went further in trying to elucidate this regime shift in the melt rate response to oceanic thermal forcing by examining the buoyancy forcing of the melt driven plume. For cold ambient temperatures the plume buoyancy is dominated by the salinity difference between the plume and the ambient water, and this salinity difference increases slowly with TF due to the increased melt water flux to the fjord. The increasing ambient water temperature however, leads to increasing temperature difference between the plume and the ambient water, leading to a negative effect on the plumes buoyancy. For sufficiently warm ambient temperatures (i.e., high TF), the negative effect due to increasing ambient water temperature difference on the buoyancy overrides the positive effect of freshening from increased input of melt water (Figure b). As a consequence, the plume velocities do not increase further with TF, resulting in effectively constant exchange coefficient in (Eq. 3) and a linear dependence of melt rates for higher TF. These results are generic and relevant for future development of the basal melt parameterizations for marine terminating glaciers in the climate ice sheet models.

*Response to subglacial discharge*

SGD, the buoyant freshwater released at depth from under Greenland's marine-terminating glaciers, is sourced largely from atmospheric-driven melting of the ice sheet surface during the summer (Chen, 2014). SGD provides an additional buoyancy source for the plume underneath the ice tongue, leading to higher basal melt rates due to higher plume velocities and entrainment of the ambient warm water (Straneo and Cenedese, 2015). Thus, submarine melting integrates both oceanic and atmospheric influences. A recent study of the relative importance of oceanic and atmospheric drivers of submarine melting at Greenland's marine-terminating glaciers from 1979 to 2018 concluded that in the north, the subglacial discharge is at least as important as variability in the oceanic thermal forcing to submarine melt rates, while it exhibits an order of magnitude larger variability on decadal time scales (Slater and Straneo, 2022). Here, we considered the response of the basal melt and melt driven circulation to varying SGD rates. In lieu of missing accurate observational estimates of SDG, we set it to be a fraction of the total basal melt for the winter control simulation. We found that the subglacial discharge (SGD) has a pronounced effect on the basal melt rates. The average melt rate for the summer control simulations (where SGD is set to 10% of the average basal melt flux for the control winter), is increased by 38%, and for the experiment with the the SGD input set to 100% of the average winter melt rate the increase in melt is 111%, consistent with the conclusions of Slater and Straneo (2022) for the northern Greenland. The additional buoyancy input affects the distribution of the melt rates and plume properties along the ice base, enhancing the melt rate and shifting the transition zone between the plume accelerating and thickening regimes closer to the grounding line. This shift of transition zone collocates with a downward thickening of the pycnocline. The functional response of the melt rate to TF found in the winter simulations (without SGD, see above) holds for the simulations with SGD, but there is stronger linear increase in the melt rate with TF for experiments with SGD as compared to the experiments without SGD. For experiments with constant TF, the melt rates increase less than linear (in a fractional manner) with the SGD, consistent with the modelling experiments of (Cai et al., 2017) for Petermann Glacier and the theoretical scaling of Slater et al. (2016).





*Future outlook*

In this work, we have focused on basal melt rates and melt driven circulation in the ice cavity under the floating tongue of Ryder Glacier, with restoring to a prescribed ocean stratification at the open boundary 30 km upstream. Future work will include the influence of sill bathymetry in the 100 km long Sherard Osborn Fjord on the oceanic heat transport to the ice cavity. Other important factors to be considered are the spatial and temporal variability of the SGD (Chen, 2014) and the three-dimensional geometry of the ice base featuring a presence of basal channels and terraces (Millgate et al., 2013; Dutrieux et al., 2014).

Including these factors in modelling studies is however contingent upon collecting accurate observational estimates necessary to initialize and evaluate the models.

*Code and data availability.* Setup files necessary to reproduce the simulations using the MITgcm, (https://github.com/MITgcm/MITgcm/releases/tag/checkpoint67s) are uploaded to the Bolin Research Centre Database under https://doi.org/10.17043/uf1nzu-1.





## Appendix A: Overview Experiments

| ExpName | $T_{AW}$ [° C] | SGD [%] | dt [s] | TF [° C] | $\tau_o$ [days] | Ave. Melt [m yr$^{-1}$] | Melt Flux [km$^3$ yr$^{-1}$] |
|---------|------|-----|-----|------|--------|----------|-----------|
| nAW20 | -2.0 | 0 | 10 | 0.68 | 78 | 0.92 | 0.16 |
| nAW15 | -1.5 | 0 | 10 | 1.18 | 44 | 3.26 | 0.55 |
| nAW10 | -1.0 | 0 | 10 | 1.68 | 34 | 6.56 | 1.12 |
| nAW05 | -0.5 | 0 | 10 | 2.18 | 30 | 10.62 | 1.80 |
| AW00 | -0.0 | 0 | 10 | 2.68 | 27 | 15.28 | 2.60 |
| **control_win** | 0.2 | 0 | 10 | 2.88 | 27 | 17.36 | 2.95 |
| AW05 | 0.5 | 0 | 10 | 3.18 | 25 | 20.48 | 3.48 |
| AW10 | 1.0 | 0 | 10 | 3.68 | 24 | 25.97 | 4.41 |
| AW15 | 1.5 | 0 | 10 | 4.18 | 24 | 31.62 | 5.38 |
| AW20 | 2.0 | 0 | 10 | 4.68 | 23 | 37.43 | 6.36 |
| AW25 | 2.5 | 0 | 10 | 5.18 | 23 | 43.40 | 7.38 |
| AW30 | 3.0 | 0 | 10 | 5.68 | 22 | 49.28 | 8.38 |
| AW35 | 3.5 | 0 | 5 | 6.18 | 22 | 55.40 | 9.42 |
| AW40 | 4.0 | 0 | 5 | 6.67 | 22 | 61.34 | 10.43 |
| AW45 | 4.5 | 0 | 5 | 7.17 | 22 | 67.31 | 11.44 |
| AW50 | 5.0 | 0 | 5 | 7.67 | 22 | 72.90 | 12.39 |
| AW55 | 5.5 | 0 | 5 | 8.17 | 22 | 78.47 | 13.34 |
| AW60 | 6.0 | 0 | 5 | 8.67 | 22 | 83.91 | 14.27 |

**Table A1.** Setup parameters and characteristic diagnostics for temperature sensitivity experiments. From left to right: Atlantic Water Temperature, subglacial discharge volume in percent of *control_win* integrated melt volume, model time step, temperature forcing, overturning time scale, averaged melt rate/ ice retreat, integrated melt (calculated for two dimensional melt).





| ExpName | $T_{AW}$ [°C] | SGD [%] | dt [s] | TF [°C] | $\tau_o$ [days] | Ave. Melt [m yr$^{-1}$] | Melt Flux [km$^3$ yr$^{-1}$] |
|---|---|---|---|---|---|---|---|
| sgd010_nAW20 | -2.0 | 10 | 5 | 0.68 | 23 | 2.37 | 0.40 |
| sgd050_nAW20 | -2.0 | 50 | 5 | 0.69 | 14 | 3.85 | 0.65 |
| sgd010_AW00 | -0.0 | 10 | 5 | 2.68 | 18 | 21.41 | 3.64 |
| sgd050_AW00 | -0.0 | 50 | 5 | 2.67 | 12 | 28.69 | 4.88 |
| **control_sum** | 0.2 | 10 | 5 | 2.88 | 18 | 23.96 | 4.07 |
| sgd020_AW02 | 0.2 | 20 | 5 | 2.88 | 15 | 26.67 | 4.54 |
| sgd030_AW02 | 0.2 | 30 | 5 | 2.88 | 14 | 28.73 | 4.89 |
| sgd040_AW02 | 0.2 | 40 | 5 | 2.87 | 13 | 30.33 | 5.16 |
| sgd050_AW02 | 0.2 | 50 | 5 | 2.87 | 12 | 31.60 | 5.38 |
| sgd070_AW02 | 0.2 | 70 | 3 | 2.87 | 11 | 33.89 | 5.76 |
| sgd100_AW02 | 0.2 | 100 | 3 | 2.86 | 10 | 36.67 | 6.24 |
| sgd010_AW20 | 2.0 | 10 | 5 | 4.67 | 17 | 47.96 | 8.16 |
| sgd020_AW20 | 2.0 | 20 | 5 | 4.67 | 15 | 52.67 | 8.96 |
| sgd050_AW20 | 2.0 | 50 | 4 | 4.66 | 12 | 60.98 | 10.37 |
| sgd010_AW40 | 4.0 | 10 | 5 | 6.67 | 16 | 76.59 | 13.03 |
| sgd020_AW40 | 4.0 | 20 | 5 | 6.66 | 15 | 83.78 | 14.25 |
| sgd050_AW40 | 4.0 | 50 | 5 | 6.65 | 12 | 96.52 | 16.42 |
| sgd010_AW60 | 6.0 | 10 | 5 | 8.67 | 16 | 103.81 | 17.66 |
| sgd020_AW60 | 6.0 | 20 | 5 | 8.65 | 15 | 114.13 | 19.41 |
| sgd050_AW60 | 6.0 | 50 | 5 | 8.63 | 12 | 130.97 | 22.28 |

**Table A2.** Setup parameters and characteristic diagnostics for subglacial discharge sensitivity experiments. From left to right: Atlantic Water Temperature, subglacial discharge volume in percent of *control_win* integrated melt volume, model time step, temperature forcing, overturning time scale, averaged melt rate/ ice retreat, integrated melt (calculated for two dimensional melt).



**Appendix B: Time series**

Time series show that for all experiments key diagnostics stabilize after 20-40 days (Figure B1 and B2). Only the integrated temperature change is increasing with time for high AW temperature experiments after an initial strong decrease (Figure B1). This increase can be attributed to a heating up of the upper layer of polar water from below. Because all other diagnostics show a statistical steady state, we can assume that the increase in heat does not influence the circulation we are investigating.

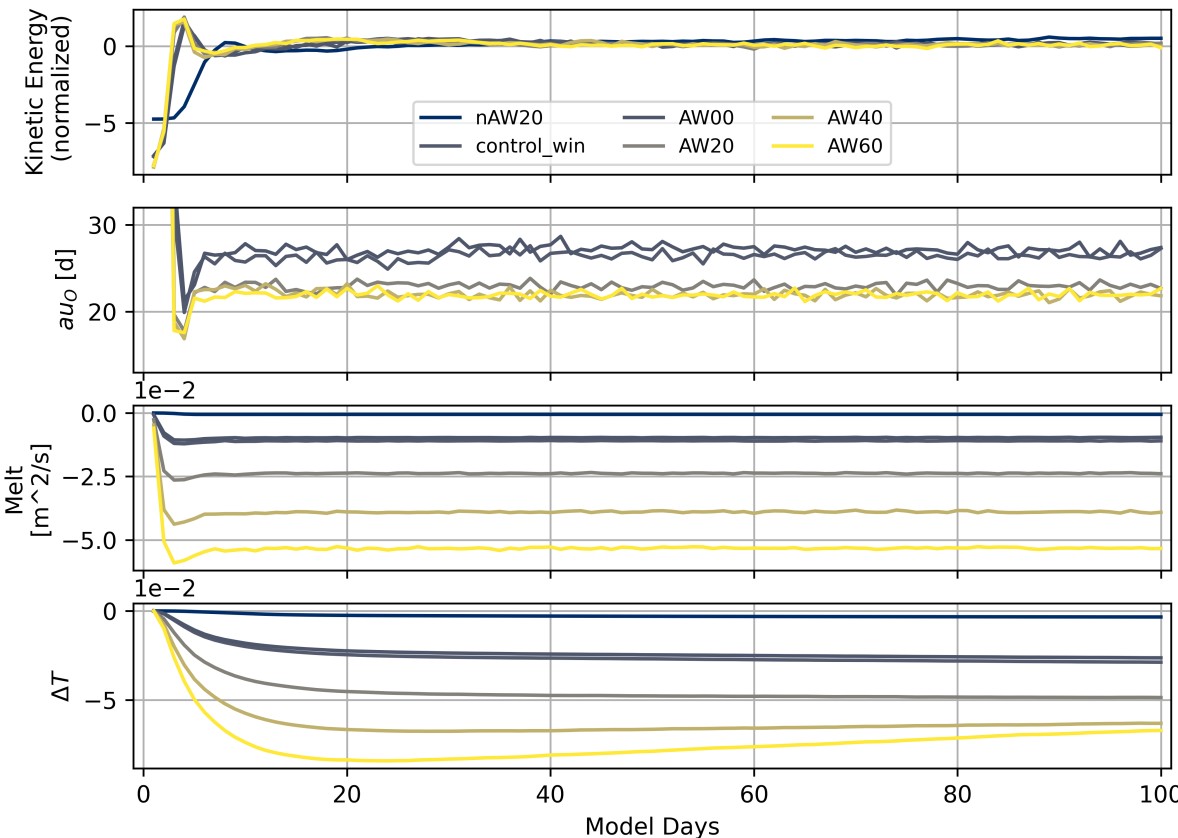

**Figure B1.** From top to bottom: Kinetic Energy, overturning timescale, melt flux (solid) and integrated temperature change (compared to initial state) as functions of model Days; shown for a representative subset of temperature sensitivity experiments.

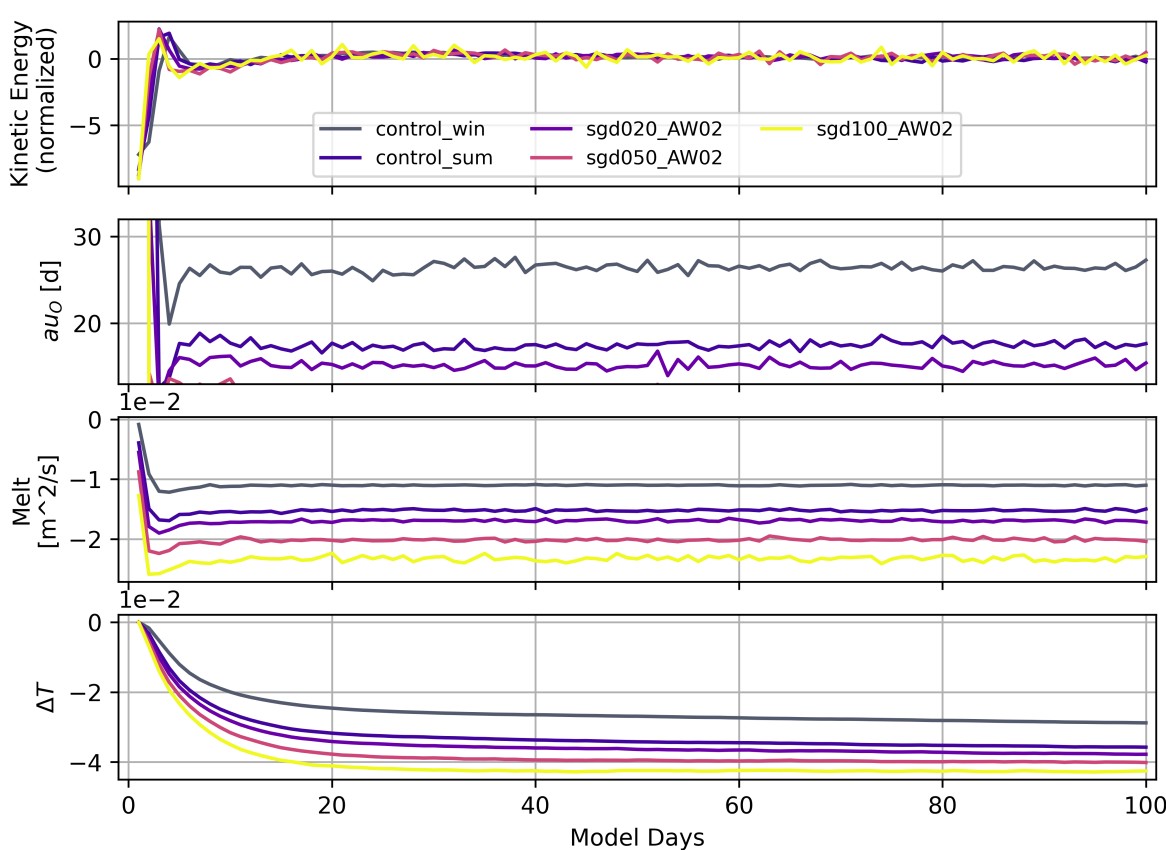

**Figure B2.** From top to bottom: Kinetic Energy, overturning timescale, melt flux (solid) and integrated temperature change (compared to initial state) as functions of model Days; shown for a representative subset of subglacial discharge sensitivity experiments.



*Author contributions.* JW conducted the model simulations and data analysis with supervision of IMK and JN. IMK and JN contributed to experiment setup and the interpretation of the results. JW wrote the first draft of the manuscript and all authors contributed to writing and editing the final version.

*Competing interests.* The authors declare that they have no conflict of interest.

*Acknowledgements.* This work was performed within a pair PhD project funded by the Faculty of Science, Stockholm University, and granted
to the Department of Mathematics, division of Computational Mathematics, and the Department of Meteorology (SUFV-1.2.1-0124-17). The computations and data analysis were enabled by resources provided by the Swedish National Infrastructure for Computing (SNIC) at the National Supercomputer Centre (NSC), partially funded by the Swedish Research Council through grant agreement no. 2018-05973. The Authors would like to thank Roberta Sciascia for fruitfull discussions.



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
