# Peer review of "Basal melt rates and ocean circulation under the Ryder Glacier ice tongue and their response to climate warming: a high resolution modelling study"

_EGUsphere, 2022_

## Referee Comment (RC1)

15/01/23
Will Scott

Title: Basal melt rates and ocean circulation under the Ryder Glacier ice tongue and their response to climate warming: a high resolution modelling study
Author(s): Jonathan Wiskandt et al.
MS No.: egusphere-2022-1296
MS type: Research article

**General comments**

I enjoyed reading the submitted paper by Wiskandt et al. They applied MITgcm to an idealised ice shelf cavity representing the ocean circulation under the Ryder Glacier tongue. This is the first time this ice shelf cavity has been modelled and this study benefits from new observations outside of temperature and salinity outside the ice tongue to force the model.

I think the authors do a good job of justifying their modelling strategy based on similar previous studies of ocean flow near tidewater glaciers. I think it would be beneficial discuss the sensitivity of grid resolution more, particularly in terms of the vertical diffusivities and vertical grid resolution as I expect that changing these parameters could have a quantitative, if not qualitative, effect on their results.

An interesting result arising from their experiments is that at higher temperature forcings the change melt rate has a linear response to temperature forcing whereas at lower temperature forcing. Similar results have been obtained for vertical ice faces at tidewater glaciers and at the ice shelf at Peterman Glacier, but I think the authors use their simulations to good effect to explain how this response appears from the simulations and how it links to previous studies of colder Antarctic ice shelves. I learnt a lot thinking about plume detachment that I am not so familiar with and this seems to be the crux of why there is a transition between the melt rate response. I might be missing something but I would appreciate more of an explanation of Figure 4b.

The results on subglacial discharge ties in with other studies in the literature. The layout and the story of the paper is very well written. This is my first review, so I would be interested to see what the other reviewers think, but I think this is a nice paper.

**Specific comments**

Line 57. Re floating ice tongues providing buttressing – not that important maybe, but should this explicitly say laterally constrained ice shelves (i.e with fjords/valleys etc). I think the point of the Gudmundsson 2013 paper was to say that if you have a laterally unconfined ice shelf it has no effect on buttressing.

Line 65. Is it worth maybe mentioning the phrase 'ice-pump' around here?

Line 74. Should this contain the Holland et al., 2008b reference, since this is one of the main papers about melt rate scaling quadratically with temperature forcing?

Line 75. I think these models were of vertical ice faces found at tide water glaciers. Is it worth mentioning that here because you've already that ice shelves are quite different to tidewater glaciers on line 55?

Line 88-90. Would it help to have another general reference for water properties in the Arctic? Maybe also for the permanent sea ice cover outside Ryder Glacier.

15/01/23
Will Scott

Line 101. The first time I read about the sills I hadn't realised these were outside the domain. (i.e outside the ice shelf). Reading it again (and looking at the pictures Jakobsson et al., 2020) it makes sense but maybe this could be more explicit. Maybe you could also add a similar sentence from your conclusion suggesting that you don't know about sills beneath the ice (if that is the case?) that might also have an effect on warm water reaching the grounding zone?

109. I think MITgcm is a finite-volume discretisation.

Line 125. Did you try varying the resolution of your model set up? In particular I imagine your vertical resolution might affect the melt rates obtained based on papers like Gwyther et al 2020. It seems like other papers that this work builds on (Xu et al 2013, Cai et al 2017) also didn't vary resolution but I think there would be more confidence in the results if you ran another a set of simulations that were 2x or 10x coarser in the horizontal/vertical and keep the parameters the same (provided the model still runs robustly) and see if it has a qualitative effect on the results.

Line 130. Could you clarify a bit more about how you got the subglacial hydrology input to work, please? Is this a standard practice in MITgcm? I was a bit surprised by the 50m grounding zone wall since this is about 15 grid cells. I thought maybe 5 would be enough… Is this similar to the methodology of Cai et al. 2017? I understand that adding sponge regions is going to make the numerics difficult. For reproducibility it might also be good to know what advection schemes you were using, i.e. types of flux limiter and general time stepping schemes.

Also I wasn't sure what the Burchard et al.,2022 reference was adding – was this an example of an enlarged grounding zone wall? Apologies if I missed that. With their paper I got the impression there were more interested in the general boundary characteristics whereas for this study, you are more interested in the dynamics themselves and how the plume starts is probably important for the resulting flow. I understand that there are lots of unknowns re the geometry and I think what you have done is fine. But I think maybe there could be a slightly more detailed explanation and tie in with the short section on sponge regions in lines 180 – 185, as well as changing the timestep in the tables. Presumably this is all part of the same problem of getting the model to run robustly? If so I think it will make it easier for people to work on this later if the details are in one section together.

136. I think using a linear eos probably is fine, but I did want to bring this up for later since there is quite a large range of temperature values that you use (at least compared to Antarctic ice shelves which I am more familiar with).

139. I think along with my point about grid resolution at line 125 – did you try varying the diffusivity and viscosity values? In particular it is the vertical diffusivity, that is going to control the vertical stratification / thickness of the plume) is. 2e-5m^2/s is probably fairly typical for an ocean model but I imagine this means you are relying on the advection scheme to add a bit of diffusion for stability. i.e if you change the grid resolution the total amount of mixing will go down because the spurious numerical mixing will be reduced.

Line 141. You say that because of turbulence you keep the diffusion and viscosity values the same (Prandtl number =1) but in fact you only have this for the horizontal viscosity and diffusivities. I think it makes sense to use similar/the same values from the previous experiments because these numbers are not well constrained and ultimately are going to depend on your grid resolution. One choice could have been to scale the vertical diffusivity/viscosity by the grid aspect ratio.

One thing that occurred to me is that for the tidewater glaciers case (which is Sciascia et al. 2013 paper) mixing away from a vertical wall is going to be strongly affected by the horizontal

viscosity/diffusivity (sciascia et al. 2013 – does actually explore this with some scaling analysis). Whereas in a shallowly sloping ice shelf the vertical diffusivity has much more of an effect on the temperature and salinity stratification. As above I think it would be interesting to know how sensitive your results are to vertical diffusivity.

Equation 3. I think I've normally seen the melt rate defined with a negative number so the melt 'wb' is a positive… It's also a bit odd to have the negative melt rate in Figure 1 but positive melt rate in Figure 3c but I can see how it fits in the plot better…

Line 160. I was a bit surprised of the form of the vertical ice shelf heat flux but looking at the MITgcm shelf ice docs maybe this is what you are doing. I think Holland and Jenkins 1999 (Section 2d) says this approximation for the ice shelf heat flux is uncommon because the ice shelf is so thick… They are other options in MITgcm so it might be worth double checking this is what you are doing.

Line 176. Just to reiterate again, I think at the end of this section is where you could add a sentence explaining that the melt paramererisation is very dependent on vertical grid resolution (and grid type) e.g see Gwyther et al. 2020.

Line 182. Is $S_b$ the salinity at the depth defined by the initial/restoring conditions? I think it is, but I think your sentence implies that it is the salinity 'after' melting which might not be the same.

Line 189. Just to reiterate I think maybe the details from Line 130 should be here/ repeated. I think expect having zero salinity is something you have to be careful with numerically because if you get negative salinity from spurious numerical mixing then the density relationships start to go wrong. I think that is why it would be useful to at least note the types of timesteppers / advection schemes used because even though it is not the focus of the paper it will help other researchers to work on similar problems later.

Another thing did you try with an open boundary at the grounding zone? I noticed in Figure 6 that you get stronger return flow at depth as the subglacial flux increases. I wonder if this would have the same pattern if the flow at the grounding zone could be balanced by the boundary condition and not recirculating flow within the cavity…

Line 192. I think this implies you changed the timestep in order to see how the melt is sensitive to timestep whereas it seems more that you were changing other parameters at the same time.

Line 224. I understand that you need a way of analysing the plume so taking u>0 seems like a reasonable choice. I am not too sure though how the resulting thickness compares to other more established plume model results. My impression was that order ~150m e.g Fig 3a, is quite large. Would you be able to find some references please to justify the magnitudes?

Line 230. The two regimes that you find is quite striking and I think the plots (e.g Figure 2 and 3) do a good job of explaining them. I am not that familiar with plume models or detachment of the plume from the ice-ocean boundary. Did you expect to see this patterns before? If so probably there should be some more references. But they are nice results.

Line 239-241. Would you be able to plot your results of temp or sal depth profile against the cruise results? If this is Figure 1d maybe it could be bigger? If that comparison is difficult maybe pointing the reader at the relevant figure in Jakobsson et al., 2020 would help.

Line 257. This is interesting! I am a bit more used to cold Antarctic ice shelves so the first question is: can AW temperature reach ~6degC ? Although you talk about their study later (and I have some more questions later), I think it is interesting that for the Holland et al 2008b study for their low

temperature cases the plume detaches. But at higher temperatures (i.e more melting, fresher plume) the plume is able to overcome the stratification and reach the top. Presumably at the lowest temperatures your plume still detaches from the ice base because the PW outside the cavity is so fresh/less dense?

Line 269. I think Figure 4a is nice and clear – there does seem to be a linear trend as TF increases and evidently that trend doesn't continues as the temperature difference gets closer to zero. Having said that, sorry I couldn't understand why you have a cut-off value which is a range between 2.88 and 3.18 degC. Could you explain your reasoning more please?

Line 270. Did you do a curve fit to find what the non-linear relationship is over the lower temperature values? I think this might make the paper stronger so you can directly compare with the work done on Greenland tidewater glaciers and ice shelfs.

Line 278. I think this is very interesting and does seem like it is the crux of the paper. I am not sure I completely understand it though. If it is obvious then just ignore the following ramble!

I can see that intuitively from Figure 4b that as you are increasing the temperature difference cooling starts to have as big an effect and balances the freshening due to melting. But it is not obvious to me that this should be the case. It's also a bit curious (and I only saw it after staring at it for a while) that it looks like the integrated plume buoyancy in Figure 4b might be starting to turn down at TF 8.68degC. Implying the cold water is having more of an effect. Did you run another simulation to check this?

Probably the first thing to say is back to my earlier point about the linear EOS, are you convinced that it is valid over these temperature ranges? Secondly, have you thought about trying to plot this using Gade lines? I wonder if you can relate the initial change in salinity directly as a function of temperature based on the gade lines. When the temp forcing is cold evidently salinity controls the density difference due to the coefficients in the eos. You might be able to plot these initial slopes on Figure 4 b to guide the eye.

Looking at the Holland et al. 2008b paper I think as part of their simple scaling analysis of the melt rate they show that the thickness of the mixed layer isn't changing. This implies that the entrainment is only a function of the velocity increase (caused by the change in buoyancy due to melting) and from there they back out the direct temperature effect on melting and the velocity effect. I wonder if you could make some plots like Figure 5 in Holland et al. 2008b because then you would be able to make / refute similar arguments that they did in section 4.

I am little bit concerned about why your plume thicknesses are so large. For instance the holland et al. 2008 mixed layer is 'only' 40m but seems fairly constant over the temperature ranges they examined (which are admittedly lower than what you have here). This might be a red-herring, but if the plume thicknesses were caused by too much spurious mixing then maybe you have a situation where the balance of melting, diffusion and temperature forcing isn't quite right. I think the results are still interesting and your results seem robust as you change the temperature forcing. That's why I made the comments earlier about the grid resolution and trade off between spurious and explicitly specified turbulent mixing. It might be worth bearing this in mind. Probably the results you have won't change but maybe the magnitudes would be different, and I think it might be worth saying that in the text. Writing this I've realised that because you have a maximum velocity/melt peak the plume has to get thicker because of convergence of the flow. I still think the approach in Holland et al 2008.b might help to work out the mechanism.

15/01/23
Will Scott

I think Jenkins 2011 might also help explain what is happening. I think in that case the buoyancy of the plume is controlled by the initial flux of subglacial discharge at the grounding zone. This is the convection driven example. Jenkins showed that in that case you actually do get linear dependence of melt rate on temperature forcing, because the plume is so fresh that adding more melt doesn't really change the density difference of the plume so the speeds are not controlled by the temperature and as a result the melt is only linearly related to temperature. It seems like you have a similar situation here although coming from a different effect, in that increasing the temperature doesn't seem to influence the buoyancy of the plume. As you point in Figure 3d and Figure 2b the density of the ambient stratification is obviously determining when the transition occurs. Maybe you could plot the integrated melt rates before 7km (in the accelerating phase) and maybe you would recover the nonlinear behaviour. Also what does the melt rate look like for nAW20. The scale you are using really doesn't show it at all... Perhaps this could be in an appendix if it helps tell the story of the two regimes.

Apologies for that long ramble but it is not clear to me (evidently!) why the integrated plume buoyancy plateaus.

Line 330-334. I am not suggesting it is a good idea but you could suggest tuning the melt rates using the turbulent exchange coefficients / drag coefficient in the melt parameterisation as I think Cai et al 2017 did this,

Line 335. Apologies if I missed it but I couldn't see in the text where there is a figure to back this up. Is this supposed to be a combination of 1d and 2b?

Line 360. I think you should probably add that Jenkins 2010 found linear response to melt rates when subglacial discharge is dominating buoyancy of plume.

Line 362. I think you should say that these references were vertical cliff tidewater glacier studies.

Line 365. Just to emphasise again I think the paper conclusions / impact would be stronger if you had calculated a curve fit for a subset of the low TF simulations to compare with the literature.

Line 367. I think 'unifying results from the previous studies' is a little bit strong maybe. Based on what you argued in the introduction that the tidewater glaciers with vertical ice fronts is different to ice shelf cavities I am not sure that you have linked ice shelves to tide water cavities. I think the main contribution which I think is still very interesting and relevant is that for high temperature differences the dynamics are quite different to Holland et al 2008b. Maybe there is a link to the Jenkins 2011 results, but I am not sure I understand it yet. They might be some caveats that this applies on small cavities where Coriolis doesn't play a role, but I think it is still very interesting. I think the same comment applies in the abstract.

Line 377. I do think the results are interesting and generic.

Line 397. I think you should cite Jenkins 2011 since that is what Cai et al 2017 and Slater et al 2016 are building on. Similarly to the nonlinear low TF, I think adding in a curve fit for the exponent would improve the paper and again make it more relevant to the existing literature.

Line 400. Maybe you could a caveat about turbulence in the conclusion. Presumably the details of the mixing / growth of the plume thickness will affect the melt rates quantitatively if not qualitatively. I think it might also be worth adding at somepoint in the discussion section about the dependence of the melt rate parameterisation on grid resolution.

15/01/23
Will Scott

**Technical corrections**

Line 22. 'and complex geometries of both, ice and ocean, domains,' I am not sure you need the commas around 'ice and ocean'

Line 51. 'in the southern Greenland' -> 'in southern Greenland'

Line 53. 'a the terminus' -> 'at the terminus'

Line 55. 'considers' seems like the wrong word. Maybe replace with 'a different type of ice-ocean interaction occurs for ice shelves'

Line 79. If there is only one other high res numerical modelling study of a Greenland ice shelf you should probably say this.

Line 96. I don't think 'rise' fits grammatically. Maybe 'suggests' instead.

Line 101. 'tide water' vs 'tidewater' consistency and in other places…

Line 114. 'nonhydrostatic' vs 'non-hydrostatic' same as above, probably best to be consistent.

Line 103. Maybe you could add more references for MITgcm used in polar settings… although you do say e.g….

Table 1. units of diffusivity and viscosity should be $m^2/s$

Table 1. I think the thermal conductivity of ice also has the units of diffusivity here based on the definition in the melt param so as above. (Holland and Jenkins 1999)

Table 1. I think you use PSU and g/kg in different parts of the text. Probably best to be consistent.

Line 163. '*' -> '+' ?

Line 163, value of C_d in table?

Line 168. 'approximated to be increase' -> 'approximated to increase'.

Line 172. Probably don't need to spell out the units.

Figure 2a. not that important but maybe you could have centralised the plot around zero velocity.

Line 248-249. You use TF before defining in the next sentence.

15/01/23
Will Scott

Line 259. I think the sentence is a little confusing. Maybe 'corresponding to' -> 'as … the plume transitions', and maybe you don't need the 'with respect to the ambient stratification'.

Line 276. 'not' -> 'no'

Figure 4a. Add residual units

Line 329. Add Ryder 'ice tongue'

Line 330-334. I found the brackets a bit confusing to understand the different values – maybe you could rephase this paragraph.

Line 347. Is this cai et al 2017 figure 2? If so might help the reader to add this in.

Line 407. I tried to access the set up files at the DOI given but the link did not work.

---

## Referee Comment (RC2)

Reviewer comments on egusphere-2022-1296:
*Basal melt rates and ocean circulation under the Ryder Glacier ice tongue and their response to climate warming: a high resolution modelling study*
J. Wiskandt, I.M. Koszalka, and J. Nilsson

**General comments**
This is a nice set of experiments on the sensitivity of an idealized Greenland ice tongue cavity to variation in ocean thermal forcing and subglacial discharge of ice sheet runoff. The results are presented in the context of other modeling studies of ice-ocean interactions in Greenland fjords and Antarctic ice shelves.

I have some comments and suggestions for revisions to the paper which fall in two main categories:
   a.  I think some of the results could be analyzed or explained more fully — particularly regarding the plume buoyancy (both its along-fjord evolution and its relationship to varying thermal forcing) — see starred comments.

   b.  References to observed/projected changes and connections to the real-world RG-SOF/GrIS systems could be expanded. This will help lend significance to the results and distinguish this paper from a more generic idealized modeling study.

One additional request: Because a large portion of the study focuses on the evolution of the plume itself, a direct comparison to a simple 1-D buoyant melt plume model simulation with the same initial T-S profiles and SGD fluxes could be quite valuable to the community in evaluating the relative benefits of running a high-resolution simulation of this nature.

I hope that these comments are useful in revising the paper and look forward to seeing this work published.

**Specific comments**
Line 82, etc. It would be great to have a map showing the RG-SOF system with the locations of the grounding line, ice tongue front, sills, and the hydrographic profiles used to initialize the model (as referenced in line 134-135), as well as maybe a smaller inset map showing the location of RG within Greenland.

Line 100. I initially thought this was saying that the sills were within the ice tongue cavity. Adding a map as suggested above would help to clarify this statement. However, I think it would make sense to move this information to the description of the model domain in Section 2.

Lines 172/210 & Figure 1a-b. In you use negative melt rates in a few places but otherwise you use positive values, which I think is more common and intuitive. This should be consistent and I would encourage you to stick to positive values = melting since you don't talk about refreezing

at all. You can still keep the way you've plotted the melt rates in Fig 1a-b by using a reversed y-axis.

Line 177. You could add a very brief intro paragraph (2-3 sentences) to Section 2.2 referencing Table 2 and A1-A2.

Lines 179-184.
1. The title of this subsection is "Oceanic thermal forcing" but then you use the term "temperature forcing" throughout the rest of the paper. I think thermal forcing is more widely used but either way, would be good to stick to one term.
2. This had me wondering (a) what typical values are for $T\_b$ in this system and (b) how $T\_{AW}$ is related to $T\_{GL}$ (i.e. is there significant mixing that occurs along the inflow pathway). From Table 2, my impression is that $T\_b$ is roughly constant at -2.68º and that the water reaching the grounding line is effectively unmodified AW. This is something you could state explicitly, i.e. TF can be estimated as $T\_{AW}+2.68º$ (as you later use in Fig 4).

Lines 185-190.
1. Could you expand a little on the values of SGD volume flux used here? I understand the general reasoning for referencing percentages of winter basal melt flux for comparison, but it would be helpful to compare the resulting values to any existing estimates of SGD volume flux (e.g. see Supporting Info S03 for Slater et al. 2022 in GRL https://doi.org/10.1029/2021GL097081 — bearing in mind that those fluxes are integrated across the grounding line while you are considering a 10 m slice, and the horizontal distribution of SGD is also likely relevant to its overall impact on basal melt, as you note elsewhere)
2. What is the vertical extent of the plume as you initialize it? Is this a typical approach to implementing SGD in this type of model?

Line 221 (& Figure 1d). It's difficult to see the differences between the simulations in figure 1d. Would it be possible to e.g. add an inset in the lower left zooming in on the lower part of the pycnocline that you reference here?

Lines 237-239. Figure 1c does not show the plume velocity dropping to zero. This made me wonder about your definition of the plume vs the outflow jet — is the outflow jet part of the plume or is it distinct? (If it's the latter you might need to refine your definition of the plume in line 223.) Does it have to do with the acceleration becoming negative? The buoyancy becoming negative?

Lines 239-241. I think by "T-S transition layer" you mean a layer of glacially-modified waters. It would be nice to see this on a T-S plot, but even without one, you could describe this more explicitly (i.e. compared to the idealized initial profiles, the outflow is colder and fresher, consistent with the signature of melt-modified ocean waters).

Lines 252-253. It took me a little while to understand what you meant by "sharpening" the pycnocline here because I was looking at the wrong part of the profile in Fig. 2b — maybe you could clarify that you're talking about "the base of the pycnocline"?

*Lines 259-260. Why is there a sudden increase in buoyancy at the regime transition? It's even more striking in the summer/SGD simulations in Fig 5d but it also happens in 3d, and this is counterintuitive to me. Is it related to the definition of buoyancy in Line 254? Since you're defining your plume using a velocity condition, is rho_p an average over the plume thickness, and rho_a is an average over the same depth range at x=21km? I'm wondering if something funky happens in that calculation as the plume reaches the pycnocline and thickens. Another possibility is that if the isopycnals are sloping significantly (hard to tell in Fig 1a-b) comparing the plume density to such a distant reference may not be ideal.

Line 269-270. Reporting the value of $T_c$ as a range between two experiments seems confusing to me (took me a while to connect this to Table A1 and understand where these values came from). I think you could simply write something like "…for experiments with a temperature forcing of $TF_c$ = 3.18ºC or greater (Figure 4a)."
In the following sentence, I initially interpreted "across the whole TF range" as including $TF<TF_c$, which isn't the case/I don't think is what you meant, so could be rephrased to clarify. It's nice to see the reduced residuals; could you also report the $R^2$ and p values for the fits here? Did you do any fitting of the range of $TF<TF_c$?

*Lines 273-281 and 368-377. This is a nice plot (4b) and interesting to think about. I think the interpretation requires a little more careful consideration here.
You've established that melt rate increases linearly with TF above ~3º. This should correspond to roughly linear decreases in plume temperature and salinity (relative to ambient). But the change in buo-T and buo-S will also depend on the changes to the ambient stratification that you have imposed. I think this is why buo-S begins to level off (while buo-T changes linearly as the melt concentration increases).
Consider that varying $T_{AW}$ while holding $S_{AW}$ and PW properties constant will change the ambient stratification and dT/dS slope. This in turn will affect the relationship between plume properties and ambient properties at a given density. I am finding this difficult to explain clearly so I'm putting a little cartoon at the end of this document in case you want to think about it more.
Do you have another explanation in mind for why the relationship between buoyancy and TF changes? Whether or not I am correct about the mechanism I think it merits a bit more thorough discussion to make clear under what conditions this result may be expected to hold.

Lines 290-292. Could the simulations with the increased $T_{AW}$ be omitted here, until the paragraph beginning line 304, to keep the structure more straightforward? Also, in the previous subsection, the simulation names from Table 2 aren't used in the text, so it would be nice to keep this consistent.

\*Lines 302-303. Just re-upping the point that while the variation in buoyancy after x=4 km or so makes sense to me, it is unclear to me why it increases abruptly around the point of the regime transition.

Lines 305-308. I think this could be expanded to at least a full sentence or two for each of these points. For point (i), it would make sense to show the regression(s) for Fig 7a to compare to the results in 4a.

Lines 308 and 397-398. Could you make a more quantitative comparison to e.g. the $x^{1/3}$ relationship found by Slater et al. (2016)?

Lines 390-392. My understanding of the Slater and Straneo (2022) paper is that it is more about the changes under realistic forcings, so this statement could be made much stronger by comparing the experiments here to observed and projected changes (see the dataset linked above in comment on lines 185-190).

**Technical corrections**
Line 128. Ice tongue terminates in a 950 m deep front, or 50 m above the sea floor (not 50 m deep)

Line 143. border (not boarder)

Line 254. Check that units here match y-axis of Fig 3d/5d?

Line 319. SGD (not SDG) — spotted in Fig 7 caption as well

Line 333. Reference Table 2.

*Figures*
Figure 1. A legend showing dashed line = summer and solid line = winter would be helpful in 1c/d.
On my screen, the dotted line in 1d looks green (not blue as stated in the caption).
What are the small blue and orange horizontal lines in 1d?

Figure 2. In last sentence of caption, could you add "The dotted horizontal lines in (b)..."

Figures 2a & 6a. It would be helpful to darken/otherwise distinguish the vertical grid line at u=0 to emphasize the change of depth of velocity reversals.

Figure 4a. Could you highlight (e.g. circle) the point corresponding to the control simulation here?

*Tables*

Table 2. Would be nice if you could further highlight the two control simulations here since the winter control is in the middle of the AW temp range (light grey shading of those rows?), and maybe add a dashed line between sgd100_AW02 and sgd010_AW20 to separate the two sets of summer experiments.

In caption, I'm not sure it's necessarily correct to imply that melt rate and ice retreat are equivalent (in your model, the ice base position is static, I think? And in reality, "retreat" would also depend on ice flow divergence?)

Re: ocean variability, ambient stratification, and plume buoyancy —

[Figure]

The slope of the Gade line can be approximated as constant. It shows that melting of ice by ocean water always creates a mixture that is colder and fresher than the water doing the melting.

Changing T_AW while holding S_AW and PW properties constant changes the slope of the mixing line between AW and PW.

In this example the cold AW (AW_c) has a shallower slope than the Gade line. The resulting mixture of AW_c and meltwater is colder and fresher than ambient water (a mixture of AW_c and PW) at the same density.

The warm AW (AW_w) has a steeper slope than the Gade line. The resulting mixture of AW_w and meltwater is warmer and saltier than ambient water at the same density — even though it is colder and fresher than the AW_w itself.

This is probably more extreme than what might be happening in the experiments here but I think the general concept might be relevant — the ambient profile is getting less stratified with a stronger temperature gradient, and the AW-PW mixing line is getting closer to the Gade slope so the salinity contrast between the plume and the ambient at a given density is getting less pronounced.

Long story short: I think it's just worth noting that the ultimate response of plume buoyancy to AW temp/TF is not straightforward and likely depends on variation in other properties as well.

Some observational context — in 79 North, AW got both warmer and saltier between 2009 and 2016, as well as over the course of the year in 2016-17, so there's reason to expect that these might covary on inter-/intra-annual timescales (https://agupubs.onlinelibrary.wiley.com/doi/10.1029/2020JC016091).
Petermann Gletscher — Washam et al. 2018 (https://journals.ametsoc.org/view/journals/phoc/48/10/jpo-d-17-0181.1.xml).
NE Greenland water mass variability — Gjelstrup et al. 2022 (https://www.nature.com/articles/s41467-022-35413-z).

---

## Author Comment (AC1)

**Author response to Reviewer 1 - Basal melt rates and ocean circulation under the Ryder Glacier ice tongue and their response to climate warming: a high resolution modelling study**

**Jonathan Wiskandt, Inga Monika Koszalka, Johan Nilsson**

**MS No.: egusphere-2022-1296**

**MS type: Research article**

Please find below our response to the comments of reviewer one to our manuscript. The reviewers comments are written in cursive, our response in regular font. Within our suggested texts, changes are marked as crossed out for deletions and in bold for additions.

**1 General comments**

I enjoyed reading the submitted paper by Wiskandt et al. They applied MITgcm to an idealised ice shelf cavity representing the ocean circulation under the Ryder Glacier tongue. This is the first time this ice shelf cavity has been modelled and this study benefits from new observations outside of temperature and salinity outside the ice tongue to force the model.

I think the authors do a good job of justifying their modelling strategy based on similar previous studies of ocean flow near tidewater glaciers. I think it would be beneficial discuss the sensitivity of grid resolution more, particularly in terms of the vertical diffusivities and vertical grid resolution as I expect that changing these parameters could have a quantitative, if not qualitative, effect on their results. An interesting result arising from their experiments is that at higher temperature forcings the change melt rate has a linear response to temperature forcing whereas at lower temperature forcing. Similar results have been obtained for vertical ice faces at tidewater glaciers and at the ice shelf at Petermann Glacier, but I think the authors use their simulations to good effect to explain how this response appears from the simulations and how it links to previous studies of colder Antarctic ice shelves. I learnt a lot thinking about plume detachment that I am not so familiar with and this seems to be the crux of why there is a transition between the melt rate response. I might be missing something but I would appreciate more of an explanation of Figure 4b.

The results on subglacial discharge ties in with other studies in the literature. The layout and the story of the paper is very well written. This is my first review, so I would be interested to see what the other reviewers think, but I think this is a nice paper.

- Thank you for your review of our manuscript and your comments—they were very helpful in clarifying several aspects of the results and improving the manuscript in general. We address the comments and detail the alterations (in boldface) in the manuscript in line with your suggestions below.

**2 Specific comments**

Line 57. Re floating ice tongues providing buttressing – not that important maybe, but should this
explicitly say laterally constrained ice shelves (i.e with fjords/valleys etc). I think the point of the
Gudmundsson 2013 paper was to say that if you have a laterally unconfined ice shelf it has no effect
on buttressing.

- Thank you for pointing this out. Our intention with citing [Gudmundsson, 2013] in the Introduction was to give a general reference for the buttressing effects of the ice shelves without delving into details as we do not resolve nor analyze these effects in detail in our study. [Gudmundsson, 2013] argues that the ice-shelf buttressing is a complex process and that the result about a laterally unconfined ice shelf having no effect on buttressing is specific to 1HD reduced version of the full Stokes system; for full Stokes system this statement is not longer valid, or at least not under all circumstances. We will clarify the citation by changing this sentence to the following:

"Under certain conditions, floating ice tongues can stabilize these glaciers by changing the stress balance and reducing the ice discharge across their grounding lines, an effect known as buttressing (Gudmundsson, 2013)."

2. Line 65. Is it worth maybe mentioning the phrase 'ice-pump' around here?

- The sentence in line 65 is meant to introduce and describe the melt driven estuarine circulation in Greenland's glacial fjords (the previous two paragraphs and following paragraphs in the Introduction focus on GrIS). We think that adding the detail about the ice-pump, i.e. refreezing, is not necessary given that there is no evidence of refreezing in our setting, it is a more common feature at colder Antarctic ice shelves. For clarity, as this sentence mentions Atlantic Water (AW) and by this refers to Greenland's fjords, we suggest instead to add a reference to Straneo and Cenedese (2015):

"The basal melt beneath the glacier ice tongue acts as a buoyancy source, driving a rising buoyant plume that forms an outflow of glacially-modified water at its neutral density level. The entrainment into the plume drives an inflow of AW towards the ice base, establishing an estuarine circulation (Straneo & Cenedese 2015)".

3. Line 74. Should this contain the Holland et al., 2008b reference, since this is one of the main papers about melt rate scaling quadratically with temperature forcing?

- Yes, that paper is a good addition here. We will add it.

4. Line 75. I think these models were of vertical ice faces found at tide water glaciers. Is it worth mentioning that here because you've already that ice shelves are quite different to tidewater glaciers on line 55?

- Correct, the cited papers that reported on a linear dependency considered the Greenland's tidewater glaciers with vertical ice fronts, which also means that they investigated warmer ocean temperatures (i.e., a stronger thermal forcing) compared to the Antarctic studies which tend to show a super-linear dependency. This is in line with our results about a linear dependence of melt rates for warmer ocean temperatures (higher thermal forcing, see Fig. 4). We will add this detail here, changing the sentence to:

"The modelling studies considered with melt rates at the Greenland's tidewater glaciers with vertical ice fronts and exposed to relatively high oceanic forcing due to warm AW, however, simulate a dependency that is not significantly different from a linear one (Xu et al., 2012; Sciascia et al., 2013)."

5. Line 88-90. Would it help to have another general reference for water properties in the Arctic? Maybe also for the permanent sea ice cover outside Ryder Glacier.

- We will add in this line a comment that the stratification found in SOF by [Jakobsson et al., 2020] is typical for the Greenlandic fjords and support it by a reference to [Straneo et al., 2012]) (which we already cited in line 43 where the Atlantic Water in the Arctic and the two-layer stratification structure is introduced):

"The hydrographic profiles show a two-layer like stratification typical of Greenlandic fjords (Straneo et al 2012), with a cold (about  $-1.5^{\circ}$  C) and relatively fresh (salinity below 34 g kg-1) surface layer (typical of Polar Surface Water, PSW) and a warm (0.2°C) and salty (34.7 g kg-1) layer of AW below 350 m."

Regarding the sea-ice cover, the reference for this information is [Jakobsson et al., 2020] (Page 3): "Available moderate resolution imaging spectroradiometer (MODIS) satellite images dating back to 2001 reveal that large icebergs, calved from the ice tongue of Ryder Glacier, remain trapped inside the fjord because the prevailing sea ice at the mouth prevents their exit." . We will add the reference to the sentence about the sea-ice cover:

"SOF is narrow ( $\sim 10$  km) rendering effects of the Earth's rotation negligible on the circulation, and a permanent sea ice cover outside of SOF inhibits wind-driven water exchange between the fjord and the open ocean [Jakobsson et al., 2020]".

6. Line 101. The first time I read about the sills I hadn't realised these were outside the domain. (i.e outside the ice shelf). Reading it again (and looking at the pictures Jakobsson et al., 2020) it makes sense but maybe this could be more explicit. Maybe you could also add a similar sentence from your conclusion suggesting that you don't know about sills beneath the ice (if that is the case?) that might also have an effect on warm water reaching the grounding zone?

- Jakobsson et al (2020) conducted a bathymetric mapping of the fjord which showed two sills located outside the cavity, there are no sills beneath the ice. We will reformulate sentences 100-101 to clarify that the study focuses on the circulation in the ice cavity excluding the sills situated outside the ice cavity, i.e. outside our model domain:

This study presents **results from a series of** high-resolution ocean-circulation model simulations of basal melt and ocean **circulation in a cavity below an ice tongue** flow in a fjord with an ice tongue. The model geometry is idealised, but its qualitative features are selected to be representative for RG and SOF. Note that SOF has two sills, which are not represented here. This is because the present focus is on flow and melt beneath the ice tongue, which are only indirectly affected by the sills: they primarily control the features of the AW reaching the ice tongue. Note that SOF has two sills outside the ice cavity, so they are not considered in model simulations presented here. The impact of the sills that control properties of AW reaching the ice cavity is a subject of a follow-up study.

- 7. Line 109. I think MITgcm is a finite-volume discretisation.
  - This is true, we will correct it.
- 8. Line 125. Did you try varying the resolution of your model set up? In particular I imagine your vertical resolution might affect the melt rates obtained based on papers like Gwyther et al 2020. It seems like other papers that this work builds on (Xu et al 2013, Cai et al 2017) also didn't vary resolution but I think there would be more confidence in the results if you ran another a set of simulations that were 2x or 10x coarser in the horizontal/vertical and keep the parameters the same (provided the model still runs robustly) and see if it has a qualitative effect on the results.

- Thank you for this very important comment. First of all, the results about the high sensitivity to vertical resolution from [Gwyther et al., 2020] are not directly transferable nor relevant to our study because they considered other ocean model families implementing different types of vertical grids and (terrain following ROMS, z-coordinate model COCO without partial cells, and MPAS-O with the terrain-following top coordinate) and much lower horizontal resolution than our MITgcm (2 km vs 10 m). Their results about sensitivity is caused by their different implementations of the ice-ocean boundary layer ("tracer sampling distance" and "flux mixing thickness", varying between these models) while the MITgcm uses the grid cell closest to the ice-ocean interface to calculate the melt rate. The high sensitivity to vertical resolution for COCO (z-coordinate) model can partly stem from its (low) horizontal resolution (see Figs. 3b and 7b). The sensitivity to diffusivity is not considered in [Gwyther et al., 2020]. Moreover, the models also followed the ISOMIP+ protocol which implies that their simulations are 3D (with spatial structure of melt rates and thermal driving in X-Y direction, Figs. 1-2). Further, horizontal pressure gradient errors specific to terrainfollowing models could influence the results. This is therefore difficult to separate the vertical resolution-dependent response from other factors present in these different models. So there is no obvious reason to expect that a z-coordinate, partial-cell MITgcm in our 2D configuration will show a similar sensitivity of the melt rates to the vertical resolution as in [Gwyther et al., 2020].

On the other hand, the sensitivity to the model resolution and viscosity/diffusivity was considered in the two MITgcm studies of similar resolution to ours, Scascia et al (2013) and in particular, by Xu et al (2012) who varied the (horizontal in their vertical plume case) model resolution by a factor of 10. Both studies concluded that while the plume got better resolved and the average melt rates increased for higher resolution, the general circulation pattern and qualitative results about e.g., dependency of melt rates on TF and SGD were consistent between the different simulations. On the other hand, the magnitude of the average melt rates was shown to depend on other factors e.g., the friction coefficient [Dansereau et al., 2013], which was used by [Cai et al., 2017] to tune the model to the observed melt rates, rather then the model resolution or diffusivity.

However, we do agree that the question about the sensitivity to the resolution is very important. Note that our configuration exhibits a relatively very high vertical (3,33 m) and horizontal (10 m) resolution compared to other ice sheet-ocean modelling studies and the values of viscosities/diffusivities are similar to Sciascia et al (2013) and at the same time as low as possible without generating numerical noise or model instability, as is often the practice [Kantha and Clayson, 2013, Cowton et al., 2015]. The values of the vertical diffusivities are moreover on the lower range of the observed in the ocean. So our study could serve as a benchmark to exploring the sensitivity to lower resolution/higher viscosity/diffusivity values, but because of the complexity of the problem we deferred it to a separate study. One important aspect is that the MITgcm is a z-coordinate model that employs partial cells while the ocean temperature and velocity values entering the parameterization are taken from the cell closest to the ice-ocean boundary. This implies that not only the vertical but also the horizontal resolution (aspect ratio of the each grid cell) will impact the melt rates, and there is an interplay between the vertical/horizontal resolution and vertical/horizontal diffusivity, which you also refer to in your later comment 11. Second, the salinity tendencies due to the melt rates are applied vertically at the immediate grid cell at the ice-ocean interface, making the response even more specific to its representation in MITgcm (compared to studies that use other definitions of the ice ocean boundary layer). Last but not least, the vertical resolution is very important in representing the ocean stratification, i.e. resolving the pycnocline between the AW and PSW layers in a Greenland fjord setting. Changing the vertical resolution and diffusivity will influence the distribution of heat and salt fluxes in the ice cavity in general as well as mixing between the plume and the ambient waters. Because all these several different aspects need to be considered and separating the different responses (sensitivities) is not trivial, a sensitivity study to the solution requires running a large number of simulations and analyses, so it deserves a separate study. We are currently planning an outline for such a study linked to another paper we are writing with colleagues from the Math Department developing a Finite Element Model for our setting, aiming towards a paper similar to [Gwyther et al., 2020] but regarding MITgcm and FEM in a Greenland fjord benchmark setting.

To sum up, the current paper is to focus on the dynamics of the response of ice shelves to oceanic thermal forcing, while the numerical aspects will be a topic of the next study mentioned above.

We will mention the planned sensitivity study in the *Future outlook* part of the Conclusions (line 402): "There are several important aspects considering the model representation of these processes. One is the sensitivity to the model resolution and viscosity/diffusivity. Previous studies using MITgcm in similar applications and resolutions, Scascia et al (2013) and in particular, by Xu et al (2012), found that while the plume got better resolved and the average melt rates increased for higher resolution, the general circulation pattern and results about the dependency on oceanic forcing and SGD were consistent between the different simulations. On the other hand, the melt rate magnitude depends also on other factors e.g., the friction coefficient [Dansereau et al., 2013], which was used by [Cai et al., 2017] to tune the model to the observed melt rates, rather then the model resolution. In our simulation with sloping ice shelf, both vertical and horizontal resolution (and viscosity/diffusivity) needs to be taken into consideration in a dedicated sensitivity study, and not only the effects on basal melt but also on the representation of the stratification and the mixing between the two water masses, AW and PSW in the domain will influence the ocean heat transport to the ice-ocean interface. Future work will also include the influence of sill bathymetry ..."

9. Line 130. Could you clarify a bit more about how you got the subglacial hydrology input to work, please? Is this a standard practice in MITgcm? I was a bit surprised by the 50m grounding zone

wall since this is about 15 grid cells. I thought maybe 5 would be enough... Is this similar to the methodology of Cai et al. 2017? I understand that adding sponge regions is going to make the numerics difficult. For reproducibility it might also be good to know what advection schemes you were using, i.e. types of flux limiter and general time stepping schemes.

Also I wasn't sure what the Burchard et al., 2022 reference was adding – was this an example of an enlarged grounding zone wall? Apologies if I missed that. With their paper I got the impression there were more interested in the general boundary characteristics whereas for this study, you are more interested in the dynamics themselves and how the plume starts is probably important for the resulting flow. I understand that there are lots of unknowns re the geometry and I think what you have done is fine. But I think maybe there could be a slightly more detailed explanation and tie in with the short section on sponge regions in lines 180 - 185, as well as changing the timestep in the tables. Presumably this is all part of the same problem of getting the model to run robustly? If so I think it will make it easier for people to work on this later if the details are in one section together.

- There are many points here. We structure the answer chronically:

Subglacial Discharge input:

Thank you for catching this - we actually had a paragraph about the SGD implementation in an earlier version of the manuscript but deleted it accidentally before the submission. We implement SGD using the RBCS package as in Sciascia et al (2013) after having discussed it with Roberta Scascia, and it is a standard method in MITgcm. As we do not have observations about the subglacial channel geometry at RG, we set the height of the subglacial channel to 20 m (same value was used by Scascia et al 2013 and Cai et al 2017). We will add these details at line 187-189 (sect. 2.2 where it fits better because it is there the sensitivity experiments to SGD are introduced):

"In lieu of lacking information about the RG's subglacial channel geometry, we assume that the subglacial flux is dispensed evenly across the grounding line in a series of ice cavities 10 m (domain across-fjord width dy) in width  $\times$  20 m in height, analogous as in 2D setups of Sciascia et al (2013) and Cai et al (2017). The subglacial flux is implemented as a source term in tracer and momentum conservation equations using MITgcm source and relaxation package RBCS (https://mitgcm.readthedocs.io/ en/latest/phys\_pkgs/rbcs.html). The subglacial discharge velocity is calculated as ratio of the SGD volume flux to the area of the model cells where the subglacial discharge is applied. Note that the subglacial discharge velocity in MITgcm is applied in horizontal direction. The SGD fluxes for various experiments are presented in Table A2. These are rescaled from the dy = 10 m wide model domain to the estimated RG grounding line width of 10 km." Please also note that the Table A2 was updated with additional SGD experiments and and the values of total SGD fluxes (instead of percentages of the melt flux) as requested by reviewer 2.

For clarity, we will also move the sentence that is currently at line 135-136: "We set up a winter control simulation  $(control_w in)$  without any subglacial discharge and a summer control simulation with subglacial discharge  $(control_s um, see sect.2.2)$ ." to the end of this section (current line 147, before sect. 2.1 "Basal melt parameterization"), so that it is clear that the details about SGD are presented later in the manuscript in sect. 2.2.

**Advection Schemes and general time stepping:**

**He asks also for time stepping. Jonathan, please check if you are using 2.14 for momentum or 2.15 and if the variables are staggered in time, see "OR":**

MITgcm uses an advective operator for momentum that is second order accurate in space, variables co-located in time and Adams-Bashforth time-stepping. In our high resolution simulation featuring stratification and strong tracer gradients, it is essential to use a flux limiting scheme (we have evaluated several advection schemes in the master thesis of Jin (2020) who used a similar SOF fjord configuration with observed T/S profiles finding that simulated flow is prone to overshoots and unrealistically low salinities when using non-conserving schemes). Here, we use the highly accurate third order direct space-time with flux limiting due to Sweby. It is explained in detail in the MITgcm documentation, including a comparison between the different tracer schemes (Section 2.17).

in https://mitgcm.readthedocs.io/en/latest/index.html). We add this information after the end of the sentence in line 142:

"The MITgm applies the semi-implicit pressure method for nonhydrostatic equations with a rigid-lid, variables co-located in time and with Adams-Bashforth time-stepping. The advective operator for momentum is second order accurate in space. We apply a third order direct space-time tracer advection scheme with flux limiter due to Sweby (https://mitgcm.readthedocs.io/en/latest/index.html, sect. 2.17)."

Grounding zone wall and Burchard 2022:

We have communicated personally with H. Burchard at the EGU Assembly last year on that issue but the reference (Burchard et al., 2022) is notadequate indeed. We take away the reference here and instead explain in more detail the choice of geometry we make:

"The grounding line is set to 50 m above the ocean floor to avoid instability issues at the corner and leave a space for the plume to develop (Burchard et al., 2022). In the absence of detailed data about the ice and sea floor topography at the grounding line we chose to keep a vertical wall below the lowest point of the ice shelf of 50 m including a 20 m vertical subglacial discharge region (970 m - 950m; see sect. 2.2) to leave place for inflowing AW and to to avoid generation of strong property gradients at the corner of the domain.

10. Line 136. I think using a linear eos probably is fine, but I did want to bring this up for later since there is quite a large range of temperature values that you use (at least compared to Antarctic ice shelves which I am more familiar with).

- Thank you for the comment, this is a good point. For lower temperature range, the difference between linear and nonlinear EOS is insignificant. At higher TF, the effect of temperature on the buoyancy of the plume would increase for a fully non-linear EOS (because the dependency of density on temperature is quadratic), which would strengthen the qualitative result about a stagnation of the buoyancy with increasing TF in figure 4b. Consistently, Scascia et al (2013) used a nonlinear EOS and also found the linear dependence of melt on TF for the (higher) temperature range they applied. We will make a note on this in the discussion (line 376):

"Note that using a fully nonlinear EOS (with a quadratic temperature dependence) instead of the linear approximation (equation 1) is unlikely to change our results about the dependency of melt on TF. At the lower ocean temperature range, the difference between a linear and nonlinear EOS is insignificant. At the AW temperatures  $> 0^{\circ}$  C, the effect of ambient ocean temperature on the plume buoyancy described above is expected to be further enhanced with a nonlinear EOS. A previous study of Sciascia et al (2013) for example, did use a nonlinear EOS and found a linear dependence of melt on TF for the AW temperatures they considered ( $0 - 8^{\circ}$  C), consistent with our result for this range."

11. Line 139. I think along with my point about grid resolution at line 125 – did you try varying the diffusivity and viscosity values? In particular it is the vertical diffusivity, that is going to control the vertical stratification / thickness of the plume) is. 2e-5m2/s is probably fairly typical for an ocean model but I imagine this means you are relying on the advection scheme to add a bit of diffusion for stability. i.e if you change the grid resolution the total amount of mixing will go down because the spurious numerical mixing will be reduced.

- Again, thanks for the suggestion. Please refer to our answer to comment 8 about a sensitivity study to model resolution and diffusivity.

12. Line 141. You say that because of turbulence you keep the diffusion and viscosity values the same (Prandtl number =1) but in fact you only have this for the horizontal viscosity and diffusivities. I think it makes sense to use similar/the same values from the previous experiments because these numbers are not well constrained and ultimately are going to depend on your grid resolution. One choice could have been to scale the vertical diffusivity/viscosity by the grid aspect ratio. One thing that occurred to me is that for the tidewater glaciers case (which is Sciascia et al. 2013 paper)

mixing away from a vertical wall is going to be strongly affected by the horizontal viscosity/diffusivity (sciascia et al. 2013 – does actually explore this with some scaling analysis). Whereas in a shallowly sloping ice shelf the vertical diffusivity has much more of an effect on the temperature and salinity stratification. As above I think it would be interesting to know how sensitive your results are to vertical diffusivity.

- Thank you for the comment. We will clarify what the values of diffusivity and viscosity are in the text (line 139-141):

"Sub grid scale processes are parameterized using a Laplacian eddy diffusion of temperature, salinity, and momentum with constant coefficients as in the MITgcm fjord simulation of comparable resolution by [Sciascia et al., 2013]. At the model resolution, the ocean mixing processes are dominated by turbulence, so we apply equal values of diffusion coefficients for all variables In the horizontal dimension we apply equal values of diffusion coefficients for temperature, salinity and momentum (horizontal Prandtl number of unity) while in the vertical the viscosity is higher than tracer diffusivity to ensure numerical stability (Table 1)."

Yes, we agree that both, vertical and horizontal resolution, will play a role and need to be investigated concurrently in a sensitivity study to follow. Please refer to our answer to comment 8 about a sensitivity study to model resolution and diffusivity. And scaling the vertical diffusivity/viscosity by the grid aspect ratio is an interesting idea to explore in the future sensitivity study, thank you for the suggestion.

13. Equation 3. I think I've normally seen the melt rate defined with a negative number so the melt 'wb' is a positive... It's also a bit odd to have the negative melt rate in Figure 1 but positive melt rate in Figure 3c but I can see how it fits in the plot better...

- For Equation 3, we adopt the formulation from [Losch, 2008], which is the reference underpinning the SHELFICE package of MITgcm we use. We will add here an explanation on how 'q' is defined (line 158):

"Equations 3 and 4, that describe heat and salt balances at the interface, respectively, are used to calculate  $S_b$  and q, where q is the upward freshwater flux (negative melt rate, in units of freshwater mass per time) and  $L_i$  is the latent heat of fusion. Upward heat flux implies basal melting (a downward freshwater flux), hence the minus sign (Losch, 2008).

While writing this comment, we noted that in this paragraph the variable  $T_i$  (ice bottom temperatures) is undefined. In fact, it should read  $T_b$  here to be consistent with equation 3 (we will correct this in line 160). We would also like to add Cai et al (2017) to the references in line 150 as they also used the SHELFICE package of Losch (2008) to simulate basal melt of the nearby Petermann Glacier.

Regarding the Figures 1a-b vs 3c and 5c, we agree that it is a bit unfortunate to plot melt rate with different sign in the different figures. Reviewer 2 suggested to flip the y axis to show positive melt rates and keep the figure compact (Figure 1). We will adopt this suggestion (see a new version of the Figure 1 below).

14. Line 160. I was a bit surprised of the form of the vertical ice shelf heat flux but looking at the MITgcm shelf ice docs maybe this is what you are doing. I think Holland and Jenkins 1999 (Section 2d) says this approximation for the ice shelf heat flux is uncommon because the ice shelf is so thick... They are other options in MITgcm so it might be worth double checking this is what you are doing.

- For this study, we decided to stick to the vertical ice shelf heat flux formulation and implementation that was used e.g., in [Cai et al., 2017] for Petermann Glacier, among other studies. We do think that it is a relevant question and we are currently working with colleagues from Grenoble on a study investigating the effect of different formulations of the vertical heat flux into the ice in MITgcm and other models. We will add the reference to Cai et al here (line 158-160):

"As in Cai et al (2017), we assume a linear temperature profile in the ice and approximating the vertical temperature gradient in the ice as the difference between the ice surface (...) divided by the local ice thickness."

---

## Author Comment (AC2)

**Author response to Reviewer 2 - Basal melt rates and ocean circulation under the Ryder Glacier ice tongue and their response to climate warming: a high resolution modelling study**

**Jonathan Wiskandt, Inga Monika Koszalka, Johan Nilsson**

**MS No.: egusphere-2022-1296**

**MS type: Research article**

Please find below our response to the comments of reviewer one to our manuscript. The reviewers comments are written in cursive, our response in regular font. Within our suggested texts, changes are marked as crossed out for deletions and in bold for additions.

**1 General comments**

This is a nice set of experiments on the sensitivity of an idealized Greenland ice tongue cavity to variation in ocean thermal forcing and subglacial discharge of ice sheet runoff. The results are presented in the context of other modeling studies of ice-ocean interactions in Greenland fjords and Antarctic ice shelves.

I have some comments and suggestions for revisions to the paper which fall in two main categories:

a. I think some of the results could be analyzed or explained more fully — particularly regarding the plume buoyancy (both its along-fjord evolution and its relationship to varying thermal forcing) — see starred comments.

b. References to observed/projected changes and connections to the real-world RG-SOF/GrIS systems could be expanded. This will help lend significance to the results and distinguish this paper from a more generic idealized modeling study.

- Thank you for your review of our manuscript and your comments—they were very helpful in clarifying several aspects of the results and improving the manuscript in general. We address the comments and detail the alterations (in boldface) with respect to the manuscript in line with your suggestions below.

One additional request: Because a large portion of the study focuses on the evolution of the plume itself, a direct comparison to a simple 1-D buoyant melt plume model simulation with the same initial T-S profiles and SGD fluxes could be quite valuable to the community in evaluating the relative benefits of running a high-resolution simulation of this nature. I hope that these comments are useful in revising the paper and look forward to seeing this work published.

- Thank you for this suggestion. A comparison between an ocean circulation model results and those from the 1-D idealized plume model from [Jenkins, 2011] is not straightforward. An ocean circulation model, like MITgcm we are using, includes for example non-linear and viscous terms and resolves the plume with several grid points in the vertical, whereas the 1-D model simulates a uniform (in the normal direction to the ice) plume. Nevertheless, we compare the resulting melt rates of both models here.

In Figure 1 we compare melt rates from our control\_sum (10% SGD) simulation with these from 1-D Jenkins plume model. The plume model is set up with the same ice geometry and the steady state temperature and salinity profile from the MITgcm simulation. We apply the same SGD flux and channel height (20 m, see section 2 in the manuscript). Entrainment and drag coefficients are taken from [Jenkins, 2011]). The MITgcm shows around three times lower melt rates than the plume model. This can be explained by higher velocities in the plume model and can be tuned by changing for example the drag coefficient or the entrainment coefficient (see e.g. [Dansereau et al., 2013, Cai et al., 2017, Slater et al., 2022]). Since the area averaged melt rates in our simulations are comparable to those from satellite observations ([Wilson et al., 2017], see discussion in the manuscript) we do not attempt the tuning.Importantly, both

Fig. 1: Simulated melt rate as a function of distance along the ice from MITgcm, the Jenkins plume model using a single layer with no stratification ("AW") and the finale profile at x=21 km from MITgcm ("Final"). Plume Model simulations are done with profiles and SGD from the control\_sum experiment from the manuscript.

models show the sensitivity to the stratification (compare "AW" and "Final" in figure 1, namely the two-regime structure in melt rates, that is described and discussed in the submitted manuscript (Section 3.1 and Discussion line 340).

**2 New Plume Criterion**

Please note, that based on the comments by the first reviewer, we adjusted our definition of the plume by adding a buoyancy criterion. This influences the values of plume thickness, velocity and average buoyancy. For a detailed discussion please see our response to reviewer one, comment 20. Updated versions of figures 3 and 5 from the manuscript are given below (Figure 4).

**3 Specific comments**

1. Line 82, etc. It would be great to have a map showing the RG-SOF system with the locations of the grounding line, ice tongue front, sills, and the hydrographic profiles used to initialize the model (as referenced in line 134-135), as well as maybe a smaller inset map showing the location of RG within Greenland.

- A map of the region is shown in [Jakobsson et al., 2020]. Because in our manuscript we focus on idealized modelling and the bathymetric results are already published in the paper above, we suggest that it is enough to add a direct reference to their figure to our manuscript: "The third largest remaining ice tongue in North Greenland belongs to the Ryder Glacier (RG) in North Greenland (54° W, 82° N, see [Jakobsson et al., 2020], Figure 1)."

2. Line 100. I initially thought this was saying that the sills were within the ice tongue cavity. Adding a map as suggested above would help to clarify this statement. However, I think it would make sense to move this information to the description of the model domain in Section 2.

- Thank you for your comment. We can clarify, that the sill are outside of the modelled domain. Therefore we also think it fits in better in the description of the area here than in the description of the model domain (as they are not in the model domain). See also the response to reviewer 1, comment 6:

"This study presents **results from a series of** high-resolution ocean-circulation model simulations of basal melt and ocean **circulation in a cavity below an ice tongue** flow in a fjord with an ice tongue. The model geometry is idealised, but its qualitative features are selected to be representative for RG and SOF. Note that SOF has two sills, which are not represented here. This is because the present focus is on flow and melt beneath the ice tongue, which are only indirectly affected by the sills: they primarily control the features of the AW reaching the ice tongue. Note that SOF has two sills outside the ice cavity, so they are not considered in the model simulations presented here. The impact of the sills that control properties of AW reaching the ice cavity is a subject of a follow-up study."

3. Lines 172/210 & Figure 1a-b. In you use negative melt rates in a few places but otherwise you use positive values, which I think is more common and intuitive. This should be consistent and I would encourage you to stick to positive values = melting since you don't talk about refreezing at all. You can still keep the way you've plotted the melt rates in Fig 1a-b by using a reversed y-axis.

- Thank you for the comment. The negative melt formulation (Equations 2-4) is taken from [Losch, 2008] as is done in [Cai et al., 2017] (see also response to Reviewer 1, comment 13). For consistency across publications (and with the model formulation) we suggest to keep the sign in the equations and add a sentence around line 172 (see response to reviewer 1 comment 13). We will adopt your suggestion of plotting positive values on a flipped y-axis (See Figure 2)

4. Line 177. You could add a very brief intro paragraph (2-3 sentences) to Section 2.2 referencing Table 2 and A1-A2.

- Thank you for your comment. We can add a short introduction about the scope of the experiments here (line 178).

"We set up two sets of experiments, one without subglacial discharge (SGD) and one including SGD. The goal of the first set of experiments is to elucidate on the dependency of basal melt on the oceanic thermal forcing. The second set is supposed to shed more light on how different SGD volumes influences the basal melt. Chosen experiments are listed in table 2. For a complete list of experiments the interested reader is referred to the appendix tables A1 and A2."

5. Lines 179-184.

1. The title of this subsection is "Oceanic thermal forcing" but then you use the term "temperature forcing" throughout the rest of the paper. I think thermal forcing is more widely used but either way, would be good to stick to one term.

- Thank you for you comment. We use "oceanic thermal forcing" when introducing or discussing the general concept of heat transport from the ocean toward the glacier (as there can also be atmospheric thermal forcing). We choose to stick to "temperature forcing" whenever we refer to our model experiments. We will carefully go through the manuscript again and make sure this is done consistently.

2. This had me wondering (a) what typical values are for  $T_b$  in this system and (b) how  $T_{AW}$  is related to  $T_{GL}$  (i.e. is there significant mixing that occurs along the inflow pathway). From Table 2, my impression is that  $T_b$  is roughly constant at -2.68° and that the water reaching the grounding line is effectively unmodified AW. This is something you could state explicitly, i.e. TF can be estimated as  $T_{AW} + 2.68^{\circ}$  (as you later use in Fig 4).

- We introduced  $T_{GL}$  mostly to exclude the possibility, that modification of the AW (due to mixing with glacially modified water or SGD water) would change the dependency of the melt on TF. As you say, the AW at the grounding line is approximately the same for all experiments (with only negligible variations or  $O(10^{-3})$ ). We can note this explicitly as suggested.

"To quantify the response of the system in terms of melt rate and circulation changes to changing oceanic thermal forcing (by varying  $T_{AW}$ ), we define an average temperature forcing (TF=  $T_{GL}(x_{GL}, z_{GL}) - T_f(x_{GL}, z_{GL})$ ) for each experiment, based on the time averaged fields when the model is in a statistical steady state (model days 61-100). where  $T_{GL}$  is the time averaged water temperature at the grounding line ( $x_{GL}, z_{GL}$ ) and  $T_f$  is the freezing point temperature evaluated at the same point using the local water salinity  $S(x_{GL}, z_{GL})$  and quantify the response of the system in terms of the melt rate and circulation changes to changing

---

## Author Response (AR1)

**Author Response**

**Review Manuscript EGUSPHERE-2022-1296**

Dear Mr De Rydt,

Thank you for your response. We are glad that you agree that our suggestions improved the manuscript.

We implemented the changes as stated in our responses to the reviewers and took into consideration the additional comments from your response:

1. We included the work by Gwyhter et al. (2020) in our discussion.
2. We added the figure and text about the comparison of our results to the 1-D plume model as an additional result in the manuscript with adjusted labels, legend and caption.
3. We added a figure showing the profiles used for the initial and boundary condition for selected experiments (not all, for better visibility) to Section 2 of the manuscript (Figure 2).

We uploaded the revised manuscript and a second document tracking all changes in the text.

Best Regards,
Jonathan Wiskandt - on behalf of the authors